# Extracellular Vesicles as Mediators of Therapy Resistance in the Breast Cancer Microenvironment

**DOI:** 10.3390/biom12010132

**Published:** 2022-01-14

**Authors:** Mark Samuels, Chiara Cilibrasi, Panagiotis Papanastasopoulos, Georgios Giamas

**Affiliations:** Department of Biochemistry and Biomedicine, School of Life Sciences, University of Sussex, Falmer, Brighton BN1 9QG, UK; cc677@sussex.ac.uk (C.C.); ppapanastasopoulos@nhs.net (P.P.)

**Keywords:** breast cancer, therapy resistance, tumour microenvironment, extracellular vesicles

## Abstract

Resistance to various therapies, including novel immunotherapies, poses a major challenge in the management of breast cancer and is the leading cause of treatment failure. Bidirectional communication between breast cancer cells and the tumour microenvironment is now known to be an important contributor to therapy resistance. Several studies have demonstrated that crosstalk with the tumour microenvironment through extracellular vesicles is an important mechanism employed by cancer cells that leads to drug resistance via changes in protein, lipid and nucleic acid cargoes. Moreover, the cargo content enables extracellular vesicles to be used as effective biomarkers for predicting response to treatments and as potential therapeutic targets. This review summarises the literature to date regarding the role of extracellular vesicles in promoting therapy resistance in breast cancer through communication with the tumour microenvironment.

## 1. Introduction

Female breast cancer (BC) is now the most frequently diagnosed form of cancer worldwide, accounting for 11.7% of all cancer diagnoses [1]. In women, BC is responsible for one in four cases of cancer and one in six cancer deaths. Nevertheless, BC is curable in around 80% of instances where early detection and the combination of loco-regional and targeted systemic approaches can render early-stage BC eradicable [2]. Screening for different molecular markers is now common practice and the results inform both treatment and prognosis. The widespread use of targeted treatments has resulted in a vastly more favourable prognosis, with a 5-year survival rate close to 90% in women with invasive, non-metastatic BC [3]. Unfortunately, recurrence and therapy resistance often occur following treatment with around 20% of patients with primary invasive BC experiencing either loco-regional or distal recurrence within 10 years [4]. Survival rates drop significantly after the onset of metastatic disease, where treatments can no longer provide a cure, but instead aim to prolong survival and reduce symptoms of the disease. This fact highlights the need for research to be conducted into the mechanisms of therapy resistance so that novel treatments can be developed to reduce the risk of recurrence and extend disease-free and overall survival.

BC treatment is heavily informed by the molecular subtype of the disease. Although historically only tumour burden was taken into account, this more rational, biologically-informed approach enables the driving characteristics of the cancer to be targeted, limiting side effects and improving treatment success. A study by Perou et al. first classified BC into four distinct subtypes: luminal A and luminal B, which overexpress the oestrogen receptor (ER), human epidermal growth factor receptor 2 (HER2)-enriched and basal-like or triple-negative [5] (Figure 1). Today, this classification has been modified to further stratify the disease based on additional molecular, histological and physical characteristics of the tumour [6]. BC positive for the hormone receptors (HRs; ER and the progesterone receptor (PR)) is generally sensitive to selective oestrogen receptor modulators (SERMs), including tamoxifen, selective oestrogen receptor downregulators (SERDs), such as fulvestrant and oestrogen withdrawal through the use of aromatase inhibitors (AIs) [7] (Figure 1). In HER2-positive BC, antibodies against the extracellular domain of the HER2 receptor (trastuzumab) or inhibitors of tyrosine kinase activity of the receptor (lapatinib) improve disease survival [8,9] (Figure 1). In both HR^+^ and HER2^+^ BC, chemotherapy and radiotherapy may be given prior to (neoadjuvant) or after (adjuvant) surgery to reduce the risk of recurrence. Triple-negative breast cancer (TNBC) is defined by the low/absent expression levels of ER, PR and HER2. In TNBC, treatment options are more limited and largely reliant on chemotherapy, radiotherapy and surgery. In recent years, the list of therapies has been growing, with some immunotherapies proving effective for TNBC. Cyclin-dependent kinase 4/6 inhibitors, which are currently approved for the treatment of advanced HR^+^/HER2^−^ BC, are showing promise in clinical trials for the treatment of TNBC [10,11] (Figure 1).

## 2. Therapy Resistance

Unfortunately, resistance to therapies is a major hurdle in the management of BC, particularly in the metastatic setting. Both intrinsic and acquired resistance are challenges that require addressing in order to improve BC treatment [12]. In intrinsic resistance, the cancer is innately unable to be targeted by a particular therapy. This can be due to genetic mutations, tumour heterogeneity or the absence of expression of the drug target. In acquired resistance, the efficacy of a treatment decreases over time until it is no longer an effective therapy for the disease. This can occur due to the activation of a parallel redundant pathway which takes over the role of the pathway targeted by the treatment that was originally driving the cancer. Moreover, alterations can occur in the drug target itself, for instance, changes in expression or mutations preventing the drug achieving its effects. Importantly, resistance can be a result of alterations in cancer cells or changes in the tumour microenvironment (TME) [13].

Although around 70% of BC is ER^+^ at diagnosis, roughly one third of women with early-stage BC treated with tamoxifen become refractory within 2–5 years [14]. Oestrogens exert their genomic action though oestrogen receptor alpha (ERα), however, in the case of endocrine therapy resistance, BC can bypass the requirement for this pathway. The activation of receptor tyrosine kinase signalling pathways, including epidermal growth factor receptor (EGFR), HER2 and insulin-like growth factor receptor (IGFR) initiates phosphatidylinositol-3-kinase (PI3K)-AKT-mammalian target of rapamycin (mTOR) signalling which can promote resistance through crosstalk with the ERα pathway. PI3K and AKT phosphorylate ERα at Ser167, activating ERα in the absence of oestrogen [15]. Additionally, mitogen-activated protein kinase (MAPK) activation can promote resistance to tamoxifen [16]. Mutations in ERα and loss of ERα expression can also lead to therapy resistance, however these changes are far less common with only around 15–20% of resistant BC losing ERα expression [14] and only 1% of BC exhibiting mutations in ESR1, suggesting that more complex signalling mechanisms are the predominant factor in hormone therapy resistance [17] (Figure 1).

HER2^+^ BC makes up 15–20% of BC and treatment with anti-HER2 monoclonal antibodies, tyrosine kinase inhibitors and antibody-drug conjugates has proven effective in increasing survival in HER2^+^ BC. Despite the utility of these treatments, resistance is a major obstacle and one study found that the addition of trastuzumab to chemotherapy only increased median survival from 20.3 months to 25.1 months in metastatic BC as resistance to treatment quickly develops [18]. Tumour heterogeneity is a strong contributor to resistance. BC cells with low levels of HER2 expression exist within HER2^+^ tumours, indicating that their growth is not driven by HER2 [19]. The selective pressure of anti-HER2 treatments leads to these cells becoming more prevalent in the tumour until the treatments are no longer effective. Loss of the extracellular domain of HER2 through cleavage by ADAM10 (disintegrin and metalloproteinase domain-containing protein 10) removes the epitope to which trastuzumab binds, rendering the treatment ineffective [20,21]. Downstream of HER2, constitutive PI3K/AKT/mTOR activation can contribute to therapy resistance by bypassing the need for HER2 activation for growth and survival [22] (Figure 1).

Chemotherapy is currently essential for the management of TNBC, however, resistance to chemotherapies is a common phenomenon and the leading cause of treatment failure in TNBC [23]. ATP-binding cassette (ABC) transporters are heavily implicated in chemotherapy resistance via the efflux of cytotoxic compounds, particularly the ABC transporter, multidrug resistance protein 1 (MDR1) [24]. Radiotherapy is another important component of many BC treatment regimens. Radio-resistance, however, can compromise the efficacy of treatment. Increased EGFR, MAPK PI3K and phospho-ribosomal protein S6 kinase beta-1 (p-S6K1), for instance, is associated with radio-resistance in BC [25] (Figure 1).

Immunotherapy is an emerging strategy in the management of BC [26]. Anti-tumour immunity arises from presentation of tumour antigens to T cells by antigen presenting cells. For example, HER2 is an important tumour-associated antigen in HER2^+^ BC [27]. Tumours quickly evolve to escape immune destruction in what is a key hallmark of cancer [28]. Immunotherapies aim to restore the anti-tumour capabilities of the immune system through vaccines, modification of immune cells and immune checkpoint inhibitors. Resistance to immunotherapy can be a result of changes in immunogenicity, mutations in antigen presentation, the formation of an immune-suppressive TME and the secretion of molecules associated with immune suppression, for instance programmed cell death-ligand 1 (PD-L1). The recruitment of specific immune cell subtypes, such as regulatory T cells (Treg), tumour-associated macrophages (TAMs) and tumour-associated neutrophils (TANs) additionally occurs, producing an immunosuppressive microenvironment. The intrinsic low mutational burden in BC compared to other solid tumours also decreases the efficacy of programmed cell death protein 1 (PD-1) inhibition [29] (Figure 1). In addition to cancer cell evolution, the TME also plays an important role in producing a treatment-resistant phenotype in BC.

## 3. The Tumour Microenvironment

As tumours grow, they form a complex network of communication with a variety of non-cancerous cell types, including innate and adaptive immune cells, fibroblasts, mesenchymal stem cells (MSCs) and endothelial cells [30]. In doing so, BCs exploit naturally existing systems to exhibit the hallmarks of cancer, enabling their survival, proliferation, invasion and metastasis, induction of angiogenesis and evasion of immune destruction [31]. The TME is carefully shaped by BC to transform it from a hostile environment for cancer growth into a cancer-promoting setting. In order to generate this transformation in the microenvironment, communication is required between BC cells and the TME. Secreted proteins, including transforming growth factor beta (TGF-β), interleukins, immunoregulatory components, oncoproteins and growth factors are major players in this transformation [32], however, the different cell types of the TME are targeted by different molecules, as described later in this review. Both local and distal cells make up the microenvironment, and distal cells are especially important in the establishment of a pre-metastatic niche [33]. In response to signals from tumour cells, changes in the TME occur, including alterations in secreted proteins, changes in motility, reduction of anti-tumour immunity and changes in vascular permeability, favouring the progression of BC [34]. In addition, certain ‘hallmarks’ are often present in the TME, as reviewed by Jin and Jin [35]. These include hypoxia, metabolic reprogramming, acidity and mechanical changes which are observed not only in BC, but in a diverse range of solid tumours. Importantly, crosstalk between BC and the TME also promotes the formation of therapy resistance [36]. Multiple paracrine and juxtacrine signalling pathways have been implicated in resistance to therapies, however, extracellular vesicles (EVs) have also been shown to contribute to therapy resistance.

## 4. Extracellular Vesicles

Extracellular vesicles are an important mediator of BC-TME signalling [37]. EVs are non-replicative, lipid bilayer-delimited particles that are naturally released from cells. They have been identified in virtually every physiological fluid and are released by nearly all cell types [38]. EV cargo consists of a number of bioactive molecules, including nucleic acids, lipids and membrane-bound and cytosolic proteins. The uptake of EVs is able to influence cell behaviour and as such, EVs are known to be important signalling particles, as well as diagnostic, predictive and prognostic biomarkers in diseases [39]. Although EVs can be categorised in a number of ways (e.g., based on size, cargo and biological role), they are most often classified based on their biogenesis, with exosomes and microvesicles being the most commonly discussed EV subtypes (Figure 2). Many other subtypes of EVs have been identified, including apoptotic bodies and oncosomes, however, knowledge of their specific roles in cell–cell communication is limited [40,41]. Due to the lack of consensus on biomarkers for specific subtypes of EVs, this review will use the collective term EV where the biogenesis pathway has not been demonstrated directly, in accordance with the guidelines set by the International Society for Extracellular Vesicles [38]. Key features of different extracellular vesicles are summarized in Table 1.

### 4.1. Exosomes

Exosomes are the most studied EV subtype, with the term ‘exosome’ often used interchangeably with ‘extracellular vesicle’ despite the lack of consensus on exosome-specific biomarkers and unavailability of methods to selectively isolate exosomes from other subtypes of EVs [38]. Exosomes are specifically defined as EVs of endosomal origin, formed through the inward budding of the membrane of late endosomes/multivesicular bodies (MVBs) to form intraluminal vesicles (ILVs), followed by the fusion of these MVBs with the plasma membrane which then releases exosomes from the cell [42] (Figure 2). The endosomal sorting complexes required for the transport (ESCRT)-dependent pathway is the main well-characterised mechanism of exosome biogenesis. It consists of five complexes which work to form ILVs in MVBs [43]. Briefly, ESCRT-0, consisting of Hrs (hepatocyte growth factor-regulated tyrosine kinase substrate) and STAM1/2 (signal transducing adaptor molecule 1/2) binds phosphatidylinositol 3-phosphate at the endosomal membrane and ubiquitin on ubiquitinated proteins, enabling ubiquitin-dependent sorting into endosomes [44,45]. ESCRT-0 recruits ESCRT-I (composed of tumour susceptibility gene 101 (Tsg101), human multivesicular body 12 (hMvb12), and the vacuolar protein sorting (Vps) proteins, Vps28 and Vps37) [46]. ESCRT-I interacts with both ESCRT-0 and ESCRT-II. ESCRT-II consists of Vps22, Vps36 and two Vps25 subunits [47]. ESCRT-II has a high affinity for phosphatidylinositol 3-phosphate, enabling localisation to endosomes. ESCRT-III is composed of the charged multivesicular body proteins (CHMPs), CHMP2, CHMP3, CHMP4 and CHMP6 [48]. ESCRT-III monomers exist in an autoinhibited state in the cytoplasm and come together upon activation and recruitment by ESCRT-II [49]. Cargo is then deubiquitinated prior to loading into ILVs and ESCRT-III dissociates from the endosome membrane by the action of ATPase Vps4 and its co-factor, vacuolar protein sorting-associated protein VTA1 homolog (VTA1) which enables the recycling of the ESCRT machinery [43]. Through this process, the endosomal membrane undergoes inward budding before scission occurs and ILVs are released into the MVB [50]. RAB proteins then mediate the transport of the MVBs to the plasma membrane along microtubules where the MVBs then dock and SNARE (soluble N-ethylmaleimide-sensitive fusion protein attachment protein receptor) complexes mediate the fusion of MVBs with the plasma membrane, releasing the ILVs as exosomes [51,52].

Additionally, exosomes can be formed in an ESCRT-independent manner. Sphingomyelinase-enriched microdomains containing ceramides within endosome membranes are thought to promote a negative curvature in the membrane, resulting in inward budding [53]. Tetraspanins also participate in cargo selection [54]. Furthermore, RAB31 has been recently shown to control an ESCRT-independent pathway where it works through flotillin proteins to selectively package cargo into ILVs and inhibits RAB7, preventing lysosomal degradation and enhancing the release of ILVs as exosomes [55].

### 4.2. Microvesicles

Microvesicles or ectosomes are formed at the plasma membrane by outward budding of the membrane followed by fission through the use of contractile machinery [56] (Figure 2). These are more heterogenous in size than exosomes. Although originally thought to be between 100 nm and 1000 nm, numerous studies have shown the existence of much smaller and larger vesicles sharing a similar biogenesis with microvesicles, for instance large oncosomes, which can reach around 10 μm in size. During microvesicle biogenesis, cargo is selected and trafficked to the membrane to be shed as microvesicles. Ras-related GTPase ADP-ribosylation factor 6 (ARF6)-positive recycling endosomes play an important role in directing cargo to the budding microvesicle. ARF6 has been found to promote the selective packaging of integrin β1, MHC-I (major histocompatibility complex class I) proteins and ARF6 itself into microvesicles [57]. The packaging of miRNA is less well-understood, however, recent evidence indicates that exportin-5 and hnRNPA2B1 play an important role. A number of other proteins participate in cargo delivery to the nascent microvesicle, including RAB22A, SNARE machinery, CD-9 and RNA-binding proteins [56]. ARF6 further promotes scission of the budding microvesicle. ARF6 promotes phospholipase D activation, recruiting ERK (extracellular signal-regulated kinase) to the membrane where it can phosphorylate myosin light chain kinase (MLCK) [58]. This then phosphorylates myosin light chain (MLC), promoting contraction around the budding microvesicle, enabling their release from the plasma membrane [57]. Moreover, RhoA works through the ROCK (rho associated coiled-coil containing protein kinase) signalling pathway to promote contractility, however, much of this pathway remains to be elucidated [59].

### 4.3. Extracellular Vesicle Separation

As with many biological components, the production of a pure EV fraction is currently unachievable. However, separation and enrichment of EVs from biological fluids can be carried out, enabling the use of EV-enriched fractions in research, diagnostics and disease monitoring [38]. The separation of EVs from conditioned cell culture media for in vitro studies and from biofluids such as plasma, urine and cerebrospinal fluid (CSF) for diagnostic purposes can be achieved in numerous ways. Among the most common methods are differential ultracentrifugation (DUC), size exclusion chromatography, ultrafiltration, precipitation and immunoaffinity-based techniques [60]. The EV fractions produced by these techniques differ in yield and purity. Importantly, they also produce EV fractions with different proportions of EV subtypes and with different levels of contaminating co-segregating components, such as apolipoproteins A1/2 and B in serum-derived EV fractions and bovine serum albumin (BSA) and secreted proteins in cell culture medium-derived EV fractions [38]. Although no method currently exists to purify different EV subtypes such as exosomes or microvesicles, this does not preclude the use of EVs in diagnostics as the EV fractions as a whole, once separated using one or a combination of these techniques, carry diagnostic, predictive and prognostic relevance in many cancers [61]. Analysis of EVs frequently includes nanoparticle tracking to determine size and quantity of EVs, transmission electron microscopy for observation of lipid bilayer vesicles and Western blotting for detection of membrane-associated and cytosolic EV markers, such as tetraspanins or integrins.

### 4.4. Extracellular Vesicle Cargo

A vast wealth of knowledge exists on the specific molecular components of EVs derived from BC cells and TME cells which are discussed in the next section. Briefly, EV cargo consists of proteins, nucleic acids, lipids and other bioactive molecules [38]. Protein cargo can include various molecules involved in signal transduction, for instance receptors and ligands. Some proteins involved in the biogenesis of the different EV subtypes are also enriched in those EVs, for instance ARF6 is found in microvesicles and ESCRT components are frequently detected in exosomes (Table 1). Additionally, immunoregulatory molecules such as immune checkpoint inhibitors and immunosuppressive cytokines can be found in EVs. Nucleic acid cargo varies between EV subtypes. Large DNA fragments have reportedly been detected in apoptotic bodies whereas smaller EV subtypes contain a more limited selection of nucleic acids, mainly consisting of miRNAs, other non-coding RNAs and mRNA [62]. At around a general length of 22 nucleotides, miRNAs are powerful post-transcriptional regulators of gene expression. Typically, miRNAs interact with the 3′ untranslated region (UTR) of mRNA to suppress translation of target genes, resulting in downregulation of proteins [63]. Often, miRNAs within EVs target genes with tumour-suppressive functions, promoting cancer progression. Furthermore, the mRNAs contained within EVs can be translated in target cells [64]. Less research has been carried out into the lipid cargo of EVs, however sphingomyelin, glycosphingolipids and phosphatidylserine are enriched in EVs [65]. Prostaglandins have also been detected in EVs and shown to contribute to signalling in recipient cells [66]. 

Changes in EV cargo during oncogenesis enable BC cells and cells of the TME to convey powerful chemical messages to surrounding cells, enhancing tumour progression and they could be used to monitor, detect and classify BC. The isolation of EVs from biofluids including, plasma, serum, urine and ascites, and the analysis of their cargo is emerging as a potential diagnostic and prognostic tool, allowing for the early detection and post-treatment surveillance of BC patients [67]. Due to their stability in biological fluids and their ability to protect and maintain the integrity of their content, preventing degradation and enabling its further study, tumour-associated EVs are currently deeply investigated through several omic techniques in order to identify novel and specific biomarkers that reflect the biological landscape of BC, the state of the tumour, disease progression and the response to cancer treatments. Specific examples of the implementation and clinical implication of these techniques are given later on in the review, highlighting how liquid biopsy, based on the non-invasive sampling and analysis of easily accessible non-solid biological tissue is a promising approach for the detection of good diagnostic, prognostic and therapeutic BC biomarkers [67].

## 5. The Roles of Extracellular Vesicles in Breast Cancer Therapy Resistance

This section discusses the roles of EVs in producing a favourable microenvironment for BC through bidirectional communication with the TME and how this promotes resistance to therapies in BC (Figure 3).

### 5.1. Fibroblasts

Fibroblasts are broadly defined as largely primitive mesenchyme-derived cells lacking lineage markers for epithelial cells, endothelial cells and leukocytes. Markers such as vimentin and platelet derived growth factor receptor α (PDGFRα) may be used in combination with cell morphology to identify fibroblasts [68]. They are located in the interstitial spaces of organs where they secrete extracellular matrix (ECM) proteins and play a central role in wound healing. During wound healing, fibroblasts can become ‘activated’ myofibroblasts and express alpha smooth muscle actin (αSMA) and TGFβ [69]. This is particularly relevant in cancer, where this transition is often exploited by BC cells due to the powerful angiogenic and immune-regulatory role that myofibroblasts can play. The formation of cancer-associated fibroblasts (CAFs) is an important event in the shaping of the TME. TGFβ, Notch signalling and inflammatory mediators have been found to be sufficient for the induction of CAF-associated proteins in fibroblasts [70]. Mechanical stress and DNA damage can also drive CAF formation, demonstrating the many pathways that can be exploited by cancer cells to induce CAF induction [71].

Dou et al. found that BC-derived EVs containing upregulated miR-146a and downregulated thioredoxin interacting protein (TXNIP) promoted the transition of fibroblasts into CAFs [72]. The induction of CAFs promotes more aggressive disease in BC through multiple mechanisms. For instance, CAFs secrete EVs enriched in miR-92 which can increase PD-L1 expression in BC, inhibiting T cell function and promoting resistance to immunotherapies through immune suppression [73]. Another study identified the protein survivin as strongly upregulated in EVs derived from the TNBC line, MDA-MB-231, after treatment with Paclitaxel. Furthermore, these EVs promoted survival when used to treat MDA-MB-231 cells, as measured by a significant decrease in death during serum-starvation. This effect was abrogated when the donor cells were treated with survivin siRNA, indicating that survivin was necessary for the promotion of cell survival through MDA-MB-231-derived EVs. Moreover, EVs from paclitaxel-treated cells decreased the sensitivity of cells to paclitaxel, an effect also mediated by survivin, as shown by the abrogation of the effect after treatment with survivin siRNA [74]. Although not directly demonstrated, the authors also suggested that these EVs may promote survival of cells of the TME. Fibroblasts are known to become activated by survivin within EVs, where they upregulate superoxide dismutase 1 (SOD1), converting them into myofibroblasts which promote BC progression and metastasis, indicating that targeting this pathway may constitute a potential therapy in BC [75].

Interestingly, fibroblast-derived EVs can also be taken up by cancer cells in the TME and sustain a more aggressive tumour phenotype. Wang et al. found that EVs derived from CAFs promoted epithelial-mesenchymal transition (EMT) in BC through miR-181d-5p [76]. The miR-181 family is known to promote both drug resistance and metastasis in BC [77]. CAF-derived EVs were shown to promote EMT in BC through the downregulation of CDX2 (caudal type homeobox 2) and HOXA5 (homeobox A5). In an MCF7 xenograft mouse model, EVs overexpressing miR-181d-5p decreased apoptosis and promoted tumour growth, whereas silencing of miR-181d-5p increased apoptosis in BC. Another study found that fibroblast-derived EVs are capable of inducing Wnt-planar cell polarity (PCP) signalling in BC cells, leading to invasiveness [78]. This was through an association of internalised EVs with Wnt11 in BC cells which activated PCP signalling. As the palmitoylation of Wnt proteins by porcupine is essential for Wnt function, the researchers found that downregulation of porcupine in TNBC cells abrogated the effects of the fibroblast EVs.

Gao et al. showed that a subset of CAFs, defined as CD63^+^ CAFs, promoted tamoxifen resistance through miR-22 [79] by downregulating ERα and PTEN (phosphatase and tensin homolog) in BC cells. The authors showed that an EV fraction separated from CAF-conditioned media using differential ultracentrifugation was capable of downregulating ERα when incubated with the ER^+^ BC cell line, T47D. Interestingly, an anti-CD63 neutralising monoclonal antibody was found to enhance the response to tamoxifen in mice compared to tamoxifen or anti-CD63 antibody alone. In a follow-up experiment, the authors produced nanoparticles containing an miR-22 sponge that could be taken up by cancer cells and reduce miR-22 concentrations. Combination of the miR-22 sponge and tamoxifen was more effective at reducing tumour growth in mouse models, suggesting that miR-22 sequestration may be an effective strategy in overcoming tamoxifen resistance produced by CD63^+^ CAFs [79].

CAFs have been shown to secrete miR-221, a microRNA able to promote endocrine therapy resistance in BC, in EVs [80]. These EVs are then taken up by BC cells, increasing Notch signalling and generating a cancer stem cell-like phenotype in BC with increased expression of CD133 [81]. Furthermore, the biogenesis of the EVs was found to be orchestrated by interleukin 6 (IL-6) signal transducer and activator of transcription 3 (STAT3) signalling [81]. This study showed that stromal cells play an important role in driving oestrogen independence in BC, contributing to endocrine therapy resistance. Another interesting study by Shah et al. identified CAFs as contributors to ER repression in ER^+^ BC cells [82]. CAFs from TNBC were able to drive the downregulation of ER in MCF7 cells through miR-221/222 present in EVs. These experiments used conditioned media to treat the BC cells, therefore more research is needed to prove this downregulation is dependent on EVs. Knockdown of miR-221/222 in the CAFs rescued this phenotype, inhibiting the EV-mediated ER downregulation [82].

An interesting study by Sansone et al. identified mitochondrial DNA (mtDNA) in circulating EVs in patients with metastatic BC that was resistant to hormone therapy. Using an MCF7 xenograft mammary fat pad mouse model, the authors found that murine mtDNA is transferred to the tumours from CAFs. This increased the oxidative phosphorylation potential of the cancers, rescuing them from depletion of their own mtDNA as a result of hormone therapy and enabling escape from metabolic dormancy. This enrichment of mtDNA, derived from EVs, was found to occur at a higher rate in cancer stem cell-like cells, compared to non-cancer stem cell-like cells. Importantly, many luminal BC cells rely on mitochondrial respiration, whereas TNBC often relies more on anaerobic glycolysis. Overall, they found that mtDNA is transferred from CAFs to BC cells through EVs and this mediated the escape of tumour cells from metabolic dormancy induced by hormone therapy [83].

Boelens et al. found that RNA within EVs derived from stromal cells stimulated STAT1-dependent anti-viral signalling through retinoic acid-inducible gene I (RIG-I) [84]. Stromal cells also activate NOTCH3, resulting in additional STAT1 activation. Initially, the authors showed that xenografting MDA-MB-231 cells with MRC5 fibroblast cells produced more radioresistant tumours than MDA-MB-231 cells alone. STAT1 expression was also lower in tumours arising from BC cells alone compared to BC with fibroblasts. Interestingly, the radioresistance was specific to basal-like BC and dependent on STAT1. This led to the expansion of a subset of therapy resistant cells through STAT1 and NOTCH3 signalling. Interestingly, inhibition of Notch reversed this resistance, improving survival in an in vivo model [84]. The same group later showed that the activation of Notch–MYC signalling in stromal cells by BC caused upregulation in the RN7SL1 (RNA component of signal recognition particle 7SL1), an RNA normally shielded by signal recognition particle 9/14 (SRP9/14). The upregulation of RN7SL1, but not SRP9/14, led to the generation of unshielded RN7SL1 which was packaged into EVs by stromal cells. The delivery of RN7SL1 to BC cells was shown to activate RIG-I, a pattern recognition receptor (PRR), increasing tumour growth, metastasis and therapy resistance [85].

### 5.2. Mesenchymal Stem Cells

MSCs are multipotent stromal cells known to contribute to tumour progression. They can be defined as CD73^+^, CD90^+^, CD105^+^ cells, lacking CD14, CD34, CD45 and human leukocyte antigen–DR isotype (HLA-DR) that can differentiate into adipocytes, chondrocytes and osteoblasts as well as exhibit plastic adherence [86]. In normal physiology, MSCs play an important role in immune responses where they act as immunomodulatory cells. In BC, MSCs enhance tumour motility and invasiveness, promoting metastatic disease through EMT. Despite this, MSCs appear to have contradictory roles in inhibiting tumour progression [87].

A study found that the treatment of adipose tissue-derived MSCs with BC-derived EVs could induce their differentiation into tumour-associated myofibroblasts, exhibiting α-SMA, stromal cell-derived factor-1 (SDF-1), vascular endothelial growth factor (VEGF), C-C motif chemokine ligand 5 (CCL5) and TGFβ expression [88]. This transition was found to be SMAD2-dependent. As mentioned in the previous subchapter, CAFs promote drug resistance through many distinct mechanisms, therefore their induction by EVs may be an important targetable mechanism in reducing therapy resistance.

Additionally, some studies reported a role of MSC-derived EVs in contributing to BC disease progression.

A comprehensive omics study by Vallabhaneli et al. found MSCs to secrete miR-21 and miR-34a in EVs, as well as a number of pro-oncogenic proteins and bioactive lipids [89]. The EVs increased survival in BC cells upon serum-deprivation, suggesting a pro-survival function. In vivo, the EVs also supported BC growth highlighting a role for MSCs in contributing to BC disease progression.

Lin et al. collected EVs from MSCs and showed that they promoted a dose-dependent increase in migration and proliferation in ER^+^ BC [90]. An analysis of gene expression following MSC-derived EV treatment revealed a number of genes were differentially expressed. In particular, Wnt/β-catenin signalling was altered. This pathway is known to contribute to drug resistance in BC, therefore, investigating the implications of this upregulation may be useful in further elucidating the roles of MSC-derived EVs in BC [91].

Furthermore, EVs from bone marrow MSC (BMMSC) cells have been shown to contain miR-23b and promote dormancy in metastatic BC cells, reducing sensitivity to chemotherapy. Indeed, disseminated BC cells can be dormant for many years before metastatic disease is realised and the slow proliferation rate of these cells renders them inherently resistant to traditional therapies that target rapidly proliferating cancer cells. Ono et al. demonstrated that BMMSC-derived miR-23b induced dormant phenotypes through the suppression of MARCKS (myristoylated alanine rich C-kinase substrate) transcripts, which encode a protein that promotes cell cycling and motility [92]. This effect was further confirmed by Bliss et al. who showed that MSCs release miR-222/223-containing EVs, promoting quiescence in disseminated BC cells [93].

Conversely to what has been reported so far, other studies highlighted a tumour-suppressor role for MSC-derived EVs, where they have been found to downregulate VEGF in BC cells, partially through miR-16 [94] and suppress angiogenesis through miR-100 [95].

### 5.3. Endothelial Cells

The outgrowth of endothelial cells is an important step in the progression of BC beyond the diffusion limit of oxygen. Initiation of angiogenesis is therefore essential for cancers to progress. Crosstalk between BC and endothelial cells promotes both vessel growth and leakiness, enabling cancer cell dissemination and metastasis, leading to treatment failure.

A group of transcription factors known as hypoxia-inducible factors (HIFs) are important players in angiogenesis in response to low levels of oxygen. The activation of these pathways promotes the expression of angiogenic genes including VEGF, angiopoietins, platelet-derived growth factor (PDGF) and fibroblast growth factors (FGFs) [96]. Although preclinical evidence supports the use of VEGF inhibition in BC, clinical studies have, so far, shown no benefit. In fact, the anti-VEGF monoclonal antibody, bevacizumab, was granted approval by the Food and Drug Administration (FDA) for BC treatment for a short time, however this was rescinded in 2011 [97,98]. The inefficacy of this treatment constitutes an innate resistance to anti-VEGF therapies, likely due to the high redundancy in angiogenic pathways, as angiopoietin-1, epidermal growth factor (EGF), FGF, PDGF and other signalling molecules can also stimulate angiogenesis. EVs have been shown to be important in the initiation of angiogenesis as well as metastasis through endothelial cell interactions [99]. Aside from BC cells, CAFs and TAMs can secrete PDGF and FGF to induce angiogenesis.

Serum EV annexin A2, which directly promotes angiogenesis, as shown by an in vivo Matrigel plug assay, correlated strongly with tumour grade and poor overall and disease-free survival in TNBC [100].

Reduction of the barrier function of endothelial cells can also be achieved by BC-derived EVs. A study found that BC neutral sphingomyelinase 2 (nSMase2) is an important regulator of miR-210 packaging into EVs, which then enhances angiogenesis upon uptake by endothelial cells [101]. Additionally, Di Modica et al. found that miR-939 in EVs from MDA-MB-231 cells reduces the barrier function of endothelial cells, enabling trans-endothelial migration in TNBC [102]. ASPH (aspartyl/asparaginyl beta-hydroxylase) expression during oncogenesis also promotes EV biogenesis in TNBC, which has been shown to enhance intravasation, enabling distal metastasis, through Notch signalling activation [103]. Furthermore, EV-mediated transfer of BC secreted miR-105, was shown to target the tight junction protein zonula occludens-1 (ZO-1), enhancing vascular permeability [104].

Interestingly, MSC-derived EVs can inhibit VEGF secretion through miR-16 and miR-100 [94,95], thereby decreasing the angiogenic potential of BC. This may constitute a useful therapeutic approach in inhibiting angiogenesis. However, the high redundancy of angiogenic pathways would suggest otherwise.

### 5.4. Adipocytes

Adipocytes are an abundant stromal cell subtype in the BC TME. Like fibroblasts, adipocytes can become cancer-associated through interactions with cancer cells, where they promote cancer development [105]. In the BC TME, cancer-associated adipocytes (CAAs) exhibit collagen VI overexpression and possess small, dispersed lipid droplets within their cytoplasm. CAAs contribute to BC proliferation through FGF and hepatocyte growth factor (HGF) secretion. Additionally, they promote angiogenesis through VEGF and IL-1β. Moreover, IL-6 and C-C motif chemokine ligand 2 (CCL2) regulate STAT3-mediated transcriptional regulation of oncogenes [106].

Wu et al. showed that BC can reprogram adipocytes to exhibit a cancer-associated phenotype through miRNAs. This led to the expression of HIF1α in adipocytes [107]. Additionally, miR-144 promoted the beige/brown differentiation of adipocytes [107].

Furthermore, adipocytes have been found to promote resistance to doxorubicin in BC through EVs. Lehuédé et al. demonstrated that the efflux of doxorubicin is mediated by the adipocyte-induced upregulation of major vault protein (MVP) in BC [108].

### 5.5. Macrophages

Macrophages are innate immune cells that act as sentinels detecting tissue damage and pathogen invasion. They can exhibit two different polarisations: the pro-inflammatory M1-type (classically activated) and the anti-inflammatory/immunosuppressive M2-type (alternatively activated) [109]. Both M1- and M2-polarised macrophages have been identified in the TME [110]. However, the immunosuppressive phenotype is predominant in the BC TME and is associated with a poor prognostic outcome, decreased relapse-free survival, and overall survival. TAMs are major contributors to malignant progression and resistance to immunotherapy [111,112,113].

Several studies have demonstrated that EV-mediated communication between macrophages, cancer cells and other cells in the TME can induce the M2 immunosuppressive phenotype.

Biswas et al. reported that MSC-derived EVs cause differentiation of monocytic myeloid-derived suppressor cells into highly immunosuppressive type 2 polarised macrophages at the tumour site [114]. Another study showed that EV lncRNA BCRT1 (breast cancer related transcript 1) derived from BC cells is internalised by macrophages to promote M2 polarisation and eventually confer increased migration and chemotaxis abilities to cancer cells, accelerating BC progression [115]. Similarly, Ham et al. revealed that the IL-6 receptor, glycoprotein 130 (gp130) packaged in BC cell-derived EVs, stimulates IL6/STAT3 signalling in bone marrow-derived macrophages (BMDMs), promoting BMDM survival and inducing the switching of BMDMs toward a cancer-promoting phenotype [116]. Furthermore, Piao et al. highlighted how macrophage polarisation, induced by BC cell-derived EVs, can create favourable conditions for lymph node metastasis in TNBC [117], while Chen et al. reported that EV miR-222 from adriamycin-resistant MCF-7 BC cells promote M2 polarisation via PTEN/AKT to induce tumour progression [118]. Moreover, Yao et al. demonstrated that BC EVs containing miR-27a-3p promote immune evasion by up-regulating PD-L1 in macrophages [119].

In addition to what is described above, many studies have also shown that specific cargos can been delivered from macrophages to BC cells to modulate tumour aggressiveness and progression. Song et al. reported that EVs containing miR-223 from macrophages can be transferred to BC cells, which result in myocyte enhancer factor 2c (Mef2c) suppression and increased invasion and metastasis [120]. Another recent study by Chen et al. revealed that HIF-1α-stabilising long noncoding RNA (HISLA), an EV-packaged lncRNA, enhances aerobic glycolysis and apoptotic resistance of BC cells via EV transmission from TAMs to tumour cells [121]. Interestingly, Yu et al. reported that EVs from macrophages exposed to post-chemotherapy apoptotic cancer cells present increased amounts of IL-6, which promote STAT3-mediated BC proliferation and metastasis [122].

On the other hand, M1 macrophage-derived EVs have been reported to sensitise BC cells to chemotherapeutics, such as carboplatin, increasing host survival in an in vivo model of BC dormancy [123]. Similarly, Moradi-Chaleshtori et al. showed that miR-33-containing BC-derived EVs can induce M1 polarisation in mice macrophages and exert an anti-tumour effect in a BC cell line [124].

### 5.6. T Lymphocytes

T lymphocytes are a component of the adaptive immune system and the second most frequent immune cell type found in human tumours besides TAMs, accounting for up to 10% of all tumour-infiltrating cells [125,126].

Many different T cell subpopulations may be found within the TME. CD8^+^ T cells and CD4^+^ T helper 1 (Th1) cells are major players of anti-tumour immune responses. Upon priming and activation by antigen presenting cells (APCs), CD8^+^ T cells differentiate into cytotoxic T lymphocytes (CTLs) and, through the release of perforin- and granzyme-containing granules, they exert a direct cytotoxic effect on target cells [127,128]. Th1 produce a high amount of proinflammatory cytokines, including interferon gamma (IFNγ, IL-2 and tumour necrosis factor alpha (TNFα which promote T cell priming and activation, CTL and natural killer (NK) cell cytotoxicity, anti-tumour macrophage activity and induction of an increase in the presentation of tumour antigens [129]. CTL and Th1 infiltration into the TME has been found to strongly correlate with a good prognosis and a longer disease-free survival [125,126]. On the other hand, Treg cells, an immunosuppressive subtype of CD4^+^ T cells, play an essential role in hindering protective immunosurveillance of neoplasia and hampering effective anti-tumour immune responses [130]. Indeed, high numbers of Treg cells in the TME correlate with poorer prognosis in BC [131].

Immunotherapy, in the form of immune checkpoint blockade, is now a therapeutic option in BC that can re-activate the cancer killing potential of T cells, improving survival amongst responders [130].

Currently, several clinical trials are investigating PD1/PD-L1 inhibitors as a monotherapy or in combination with other therapies, such as chemotherapy, radiotherapy, poly adenosine ribose polymerase (PARP) inhibitors, or angiogenesis inhibitors in patients with metastatic TNBC, which is well-established as being the most immunogenic BC subtype [130]. Based on the results of the IMpassion 130 study, the anti-PD-L1 drug atezolizumab plus nab-paclitaxel was approved from the FDA for unresectable locally advanced or metastatic TNBC with PD-L1 expression [132]. However, while patients generally respond well to immunotherapy, a certain proportion of them present resistance to this therapeutic strategy [130].

Increasing evidence indicates that EVs can mediate the abnormal expression of immune-checkpoint proteins and eventually induce the onset of therapeutic resistance. Several studies have shown that BC-derived EVs containing PD-L1 can induce an enhanced immunosuppressive phenotype in BC, impairing the activation and cancer killing potential of T cells and enhancing tumour growth [133,134,135]. Moreover, TGFβ has been identified as a promoter of PD-L1 packaging in BC-derived EVs [136,137], and Poggio et al. demonstrated EV PD-L1 resistance to anti-PD-L1 blockade [138].

Interestingly, other than BC-derived EVs, CAF-derived EVs can also regulate PD-L1 expression and mediate therapeutic resistance. It has been shown that after treatment with CAF-derived EVs, BC cells express higher PD-L1, which significantly impairs T cell proliferation and anti-tumour activity [73].

Beyond their role in the immune checkpoint regulation, EVs have also been described for their wider ability in modulating or reprogramming the functions of T cells, cooperating in the building of an immunosuppressive TME, which eventually mediates immune escape and therapeutic resistance. Wen et al. showed that EVs derived from highly metastatic BC cells directly suppressed T cell proliferation and inhibited NK activity and hence likely suppressed the anti-cancer immune response in pre-metastatic organs [139]. In addition, it is now clear that BC-derived EVs can also induce and sustain Treg cells in the TME. A study by Ni et al. showed that CD73^+^γδ1 Treg cells were induced through EV-mediated delivery of the lncRNA SNHG16 (small nucleolar RNA host gene 16). In the recipient cells, lncRNA SNHG16 competitively bound to miR-16-5p, enabling the activation of the TGFβ1 SMAD5 pathway and promoting the expression of CD73 [140]. Moreover, Ning et al. demonstrated that BC-derived EVs distinctly inhibited CD4^+^IFNγ^+^ Th1 differentiation but increased the rates of Treg cells [141].

Furthermore, an interesting recent paper by Schwich et al. presented a novel EV-mediated mechanism of immune escape. A pronounced immunosuppressive/exhausted phenotype was achieved through the interaction of the human leukocyte antigen-G (HLA-G), secreted in the blood via BC-derived EVs, and the inhibitory receptor immunoglobulin-like transcript 2 (ILT-2), expressed on CD8^+^ T cells [142].

### 5.7. Dendritic Cells

Dendritic cells (DCs) are APCs with the main role of presenting antigens to T cells in order to initiate an immune response [143]. They are often thought of as the link between innate and adaptive immunity due to their role in detecting dangers and presenting antigen material to T cells, stimulating the adaptive arm of the immune system [144]. In cancers, DCs are often immature, without the ability to present antigens to T cells. DCs can additionally be proangiogenic and immune-suppressive, inhibiting T cell proliferation [145].

Interestingly, in contrast to many of the pro-tumour effects mediated by EVs, DCs have been shown to utilise EVs in producing an anti-tumour phenotype in CD8^+^ T cells. Anti-tumour T cell activity is reliant on the presentation of tumour antigens on DC MHC-I molecules. Wolfers et al. showed that tumour cell-derived EVs can be taken up by DCs, transferring tumour antigens and enabling the initiation of anti-tumour immunity in CD8^+^ T cells [146]. As a result, tumours employ a number of mechanisms to evade this immune detection. One study found that tumour-derived EVs contained fatty acids that activate peroxisome proliferation activated receptor alpha (PPARα) in DCs. This led to the enhancement of immune evasion through altered fatty acid metabolism. The effect was also reversed by PPARα inhibition which restored the function of DCs, enhancing the efficacy of immunotherapies [147].

Another study noted that BC-derived EVs were enriched in double-stranded DNA (dsDNA) following irradiation. Radiotherapy produces cytosolic dsDNA in cancer cells and Diamond et al. found that it was packaged into EVs and transferred to DCs [148]. This led to IFN-I activation via the stimulator of interferon genes (STING) pathway [149]. Once inside DCs, the dsDNA promoted anti-tumour immunity through CD8^+^ T cell responses. This was shown in an in vivo mouse model. This is not the only study to show the anti-tumour effect of DC-derived EVs, with another demonstrating that murine tumours were eradicable following the treatment of mice with DC-derived EVs [150].

### 5.8. Myeloid-Derived Suppressor Cells

Myeloid-derived suppressor cells (MDSCs) are strongly immunosuppressive CD33^+^, CD11b^+^, HLA-DR^−^ myeloid lineage cells [151]. In BC, MDSCs are present at a much higher level than in healthy individuals and their abundance in cancer tissues is associated with poorer prognosis and therapy resistance [152].

A recent study identified MDSC-derived EVs as key mediators of growth, invasion, angiogenesis and immunosuppression. The latter was found to occur through the recruitment of immunosuppressive cells, limiting T cell cytotoxic function by IL-12, IL-13, IL-1Ra, IL-4, LIX (liposaccharide-induced CXC chemokine) and TNFα, found in MDSC-derived EVs. In an in vivo experiment, the administration of these EVs to mice resulted in a decreased CD8^+^ T cell and M1 macrophage population, while increasing the abundance of M2 macrophages [153]. Another study showed that EVs derived from BC are capable of inducing bone marrow cells to differentiate into MDSCs [154], promoting an immunosuppressive microenvironment through the suppression of T cells.

Moreover, some studies highlighted the EV-mediated ability of MDSCs to favour the formation of a pre-metastatic niche, enabling progression of BC into an incurable metastatic disease. Wen et al. found that EVs from highly invasive BCs were more capable of inducing the accumulation of CD11b^+^/Ly6Cmed gMDSCs, immature myeloid cells whose abundance has been linked to poorer prognosis in cancer, in the lungs and livers of mice than non-metastatic BC. [139]. Another study found that doxorubicin stimulated BC cells to induce IL-13^+^ Th2 miR-126a^+^ MDSCs which promoted BC lung metastasis through the release of EVs containing miR-126a [155]. Gu et al. also reported that EVs can regulate distal organ metastasis in BC. They found that CCL2 in lung tissue promotes the recruitment of MDSCs, enabling the establishment of a pre-metastatic niche. In a mouse model, EVs were taken up by alveolar epithelial type II cells. The EV cargo responsible was identified to be miR-200b-3p that targets PTEN [156].

### 5.9. Natural Killer Cells

NK cells are important lymphocytes with roles in mediating anti-tumour immunity [157]. Their ability to autonomously kill tumour cells suggests that they have potential to be used in anti-tumour immunotherapies [158]. However, under physiological conditions, NK cells quickly become unable to suppress BC growth as the cancer evades detection and destruction.

Although research into EV-mediated BC-NK interaction is limited, one study from Wen et al. found that murine BC cell-derived EVs leads to the accumulation of MDSCs in the lungs and liver, suppressing NK cell cytotoxicity and eventually enabling the conditioning of the pre-metastatic niche [139].

Additionally, Zhang et al. discovered that the EV fraction derived from BC cells can inhibit IL-2-induced NK cell cytotoxicity. Interestingly, they also found that curcumin reverses this effect, increasing NK cell activity in mice [159].

### 5.10. Neutrophils

Neutrophils are the most abundant circulating leukocytes with essential roles in innate immunity, working through phagocytosis, the release of granules and the use of neutrophil extracellular traps (NETs) [160]. In cancer, neutrophils are more abundant and their number correlates with disease progression [161]. Although neutrophils have short lives, their role in inflammation is sufficient to contribute to malignancy through multiple mechanisms [162]. Interestingly, as with macrophages, neutrophils can possess two different phenotypes in cancer, namely N1 (anti-tumour) and N2 (pro-tumour) tumour-associated neutrophils [163].

Surprisingly, very little research has been carried out into how neutrophils and BC cells interact through EVs. One study identified BC-derived EVs as contributing to higher levels of neutrophils in mice injected with BC cancer cells. Additionally, BC-containing mice more readily formed thrombi due to NET formation. EVs themselves induced this effect as their injection into granulocyte colony-stimulating factor (G-CSF)-treated mice accelerated thrombus formation [164].

## 6. Clinical Applications of Extracellular Vesicles in Therapy Resistance

Mechanistically, EVs have been strongly implicated in the development of therapy resistance in BC, therefore, several attempts have been made to use this information in clinical settings. This section discusses how EVs can be used as biomarkers in BC as well as how they may be used in treatments, both as drug targets and as vehicles of cargo delivery.

### 6.1. Extracellular Vesicles as Biomarkers

As circulating EVs and their contents may represent an accurate ‘snapshot’ of the current status of the underlying molecular processes within cancer cells, but also of the interactions between cancer cells and the supporting TME, EV ‘liquid biopsies’ can potentially provide an easily accessible diagnostic tool in BC, but may additionally carry prognostic and predictive information that can inform clinical treatment decisions. Below we summarise the important recent developments focusing on the role of circulating EVs that highlights their importance in BC diagnosis and prognosis (Table 2).

One study isolated and identified phosphoproteins extracted from EVs from human plasma as potential markers to differentiate disease from healthy volunteers. They identified close to 10,000 unique phosphopeptides in EVs isolated from small volumes of plasma samples. Using label-free quantitative phosphoproteomics, the authors identified 144 phosphoproteins in plasma EVs that were significantly higher in patients diagnosed with BC compared to healthy controls. This study shows that the development of phosphoproteins in plasma EVs as disease biomarkers is highly feasible and may potentially transform cancer screening and monitoring [165].

Furthermore, profiling the three tumour-associated protein markers: EGFR, epithelial cell adhesion molecule (EpCAM) and HER2 on circulating EVs in a BC patient cohort, resulted in diagnosing breast tumours with high efficiency (AUC: 0.9845) (area under the receiver operating characteristic curve) and a high sensitivity of 97.37% for distinguishing malignant BC vs. healthy controls, whereas very early, stage I cases were detected with 92.31% sensitivity [166]. Similarly, in a different study, the percentage of CD63/EpCAM/mucin 1-triple-positive circulating EVs in BC patients was significantly higher than that of healthy controls and this was associated with an overall accuracy of 91% in BC diagnosis [167]. In a similar study where a microfluidic chip was used for immunocapture and quantification of circulating EVs from patients with BC, a significant increase in the EpCAM-positive EV expression level was detected compared to healthy controls, whereas EV HER2 expression levels were almost consistent with that in tumour tissues assessed by immunohistochemical staining [168]. This study provides further evidence to support a potential diagnostic role for circulating EVs in BC, but also a role in disease sub-classification on the basis of a liquid biopsy.

Another very recent study aimed to investigate the morphology and phenotype of EVs derived from early-stage BC patients. Patient plasma samples were taken before surgery, and 1 and 6 months after surgery, throughout adjuvant therapy, to investigate the value of EVs as cancer markers in this clinical setting during the earliest stage of the disease, thus aiming to identify possible biomarkers to monitor patients in the first months after surgery and/or during adjuvant therapy. In vesicles derived from BC patients among three time points, a significant time effect was detected in EV mean diameters, in particular for pre-surgery vs. post-surgery (median EV mean diameter 131.1 nm vs. 142.4 nm, *p* = 0.021) and for 1 month post-surgery vs. 6 months post-surgery (median EV mean diameter 142.4 nm vs. 113.2 nm, *p* = 0.02). However, EVs derived from BC patients did not show a variation in diameter compared to those from healthy subjects. Plasma EV analysis showed that 11 significant markers were able to significantly discriminate between healthy subjects and patients: CD3, CD56, CD2, CD25, CD9, CD44, CD326, CD133/1, CD142, CD45, and CD14. All markers significantly distinguished healthy subjects and BC cases: CD3, CD25, CD56 (*p* < 0.001); CD2, CD9, CD142, and CD14 (*p* < 0.01); CD44, CD326, CD133/1, and CD45 (*p* < 0.05). Statistical results confirmed the trend of tumour samples to have, on average, higher marker values than healthy samples, except for CD45 which decreases its fluorescent intensity in BC cases. Statistical differences were further observed within different time points of BC patients for CD146 (*p* = 0.034) and CD45 (*p* = 0.047). More specifically, both markers were found to be downregulated 1 month after surgery compared to baseline measurements (CD146 *p* = 0.042 and CD45 *p* = 0.040), suggesting that these markers can further have a value for monitoring disease after surgical resection, during adjuvant therapy and in the event of disease recurrence [169].

In another study, circulating EVs isolated from the plasma of 10 patients with BC (stages I and II) and 5 healthy controls were analysed using LC-MS/MS (liquid chromatography with tandem mass spectrometry). Developmental endothelial locus-1 protein (Del-1) was selected as a cancer-specific candidate biomarker. Circulating EV Del-1 levels were significantly higher (*p* < 0.0001) in patients with BC compared to healthy controls and returned to almost normal after tumour removal. The diagnostic accuracy of Del-1 was AUC 0.961 [95% confidence interval (CI), 0.924–0.983], sensitivity of 94.70%, and specificity of 86.36% in the test cohort and 0.968 (0.933–0.988), 92.31%, and 86.62% in a validation cohort for early-stage BC [170]. A different study focused on identifying novel epithelial markers in circulating EVs through the development of a dual sandwich-type electrochemical paper-based immunosensor for claudin-7 and CD81 determination. The authors validated claudin-7 expression levels in EVs from 60 patients with early-stage BC and 20 healthy volunteers and the results showed that the levels of claudin-7 are significantly higher in patients with BC than in healthy controls, with an AUC of 0.8517 ± 0.06 SD (standard deviations) to distinguish the two groups. Furthermore, using CD81 as a housekeeping protein, the claudin-7/CD81 ratio to distinguish patients with BC from healthy controls was found to improve the accuracy of claudin-7 in the diagnosis of BC (AUC = 0.8908 ± 0.048 SD). Both claudin-7 and claudin-7/CD81 ratios were more accurate as diagnostic markers than conventional serum markers carcinoembryonic antigen (AUC, 0.5217 ± 0.068 SD) and mucin 1 (AUC, 0.7683 ± 0.0056 SD) [171].

Furthermore, a study of EVs isolated from the urine of patients with BC and healthy controls demonstrated that the combined expression of miR-21 and matrix metalloproteinase-1 (MMP-1) in urinary EVs detects 95% of BC without metastasis. More specifically, miR-21 expression in the BC patients was significantly lower than in the 26 controls, whereas MMP-1 expression in patients was significantly higher than in controls, thus miR-21 and MMP-1 may be useful markers for BC screening in urine samples [172].

In a study of quantified levels of the lncRNA, H19 in serum-derived EVs from patients with BC or benign breast disease and healthy subjects, using quantitative real-time PCR, EV H19 expression levels were upregulated in patients with BC compared to that in patients with benign disease and healthy controls. Furthermore, the median serum EV H19 levels were significantly lower in post-operative settings than that in pre-operative settings, whereas EV H19 expression levels were associated with lymph node metastasis, distant metastasis, TNM stages, ER, PR, and HER2 expression, results indicating that serum EV H19 may act as a novel biomarker for the diagnosis and monitoring of BC [173].

Moreover, in another recent study, the expression of EV miR-1910-3p was significantly higher in the serum of BC patients than in that of healthy volunteers. The diagnostic sensitivity, specificity, and area under receiver operating characteristic (ROC) curve for miR-1910-3p in serum EVs were 88%, 76%, and 0.8864, respectively. The diagnostic sensitivity, specificity, and area under ROC curve for serum CA15-3 were 68%, 60%, and 0.6624 respectively, indicating that the diagnostic efficacy of serum EVs miR1910-3p was significantly higher than that of the traditional tumour marker CA15-3. A sensitivity of 98% was achieved when these two tumour markers were used in combination for the diagnosis of BC, which was significantly higher than the sensitivity of either one of the two markers alone. Therefore, the combined use of miR-1910-3p in serum EVs and serum CA153 can significantly improve the sensitivity of BC diagnosis and may serve as a new molecular detection method for BC [174].

In a similar study of circulating EV miRNAs in patients with BC and healthy controls, it was shown that EV miRNA-21 and 105 expression levels were higher in metastatic versus non-metastatic patients and healthy controls, whereas higher levels of miRNA-222 were observed in basal-like and in luminal B versus luminal A tumour subtypes. EV miRNA-222 levels correlated with clinical and pathological variables such as progesterone receptor status and Ki67. During neoadjuvant treatment, EV miRNA-21 expression levels directly correlated with tumour size and inversely with Ki67 expression. Finally, higher levels of EV miRNA-21, miRNA-222 and miRNA-155 were significantly associated with the EVs of circulating tumour cells. In conclusion, this study demonstrated that liquid biopsies based on EV miRNAs and circulating tumour cells can be a complementary clinical tool for improving BC diagnosis, prognosis and phenotypic subclassification [175].

In a recent proteomic analysis of circulating EVs in BC patients, profiling of cancer-associated proteins from plasma EVs without the interference of soluble proteins resulted in a EV protein signature (a weighted sum of eight EV protein markers; carcinoma antigen 15-3 (Ca15-3)), carcinoembryonic antigen (CEA), cancer antigen 125 (Ca125), HER2, EGFR, prostate-specific membrane antigen (PSMA), EpCAM and VEGF with a high accuracy (91.1%) for discrimination between metastatic BC, non-metastatic BC and healthy controls. Within the same study, in a prospective cohort of 35 plasma samples, the EV signature achieved an accuracy of 85.2% for discrimination between progressive disease and partial response/stable disease and showed similar performance for classifying treatment response when applied to different BC subtypes, HR^+^, HER2^+^ and TNBC. Moreover, the authors compared the performance of the EV signature and plasma CA 15-3 in monitoring response to systemic treatments in metastatic BC patients. The change in tumour burden could be better captured by the EV signature than plasma CA 15-3 across different BC subtypes. HR^+^ metastatic BC patients with continuous partial response showed a decreasing level of the EV signature, as compared to a slight increase in the concentration of plasma CA 15-3. For HER2^+^ metastatic BC patients and metastatic TNBC patients showing disease progression, the levels of EV signature were elevated. However, the concentration of plasma CA 15-3 remained unchanged or even decreased at the time of disease progression. These results reveal that the EV signature could be used for longitudinal monitoring of therapeutic responses. Furthermore, the performance of EV protein profiles in predicting clinical outcomes was investigated in a cohort of 59 metastatic BC patients who were undergoing therapies and had baseline EV protein profiles available. A high level (above median value) of the EV signature was significantly associated with inferior progression-free survival (PFS) in Kaplan–Meier analysis (log-rank test: *p* = 0.028). Median PFS was 475 days for low values of EV signature, as compared to the median PFS of 254 days for high values of EV signature. Cox proportional-hazard regression analyses using a univariate model revealed that the EV signature was a strong predictor (hazard ratio (HR) = 4.1, 95% CI= 1.1–16.4, *p* = 0.0405) of PFS in metastatic BC. Moreover, the EV signature remained an independent predictor (HR= 6.4, 95% CI= 1.5–27.4, *p* =0.0129) in multivariate analysis when adjusting for age and immunohistochemical status of ER, Ki67, and HER2. In contrast, plasma CA 15-3 did not show prognostic value in the same cohort. To validate the performance of the EV signature in the prognostic prediction of PFS, the authors further collected 16 plasma samples from metastatic BC patients prior to treatment. This prospective cohort verified that a higher value of EV signature was significantly associated with an inferior PFS in Kaplan–Meier analysis (log-rank test: *p* = 0.033 for the EV). Collectively, these results imply that the EV signature may serve as an independent marker for metastatic BC prognosis [39].

A recent study aiming to examine the expression and function of EV annexin A2 (exo-AnxA2) derived from the serum samples of BC patients showed that the expression of serum exo-AnxA2 in BC patients was high compared to healthy females. High expression of exo-AnxA2 levels in BC patients was significantly associated with tumour grade (*p* < 0.0001), poor overall survival (hazard ratio (HR) 2.802; 95% confidence intervals (CI) = 1.030–7.620; *p* = 0.0353) and poor disease-free survival (HR 7.934; 95% CI = 1.778–35.398; *p* = 0.0301). The expression of serum exo-AnxA2 levels was significantly elevated in TNBC in comparison to ER^+^, HER2^+^ and healthy females [100].

Recently, a comparative proteomic analysis of plasma EVs was performed in healthy controls and BC patients undergoing chemotherapy, radiotherapy or after surgery; proteomic analysis of sEV-enriched fractions using a reverse phase protein array revealed a signature of seven proteins that differentiated BC patients from healthy individuals, of which, focal adhesion kinase (FAK) and fibronectin displayed high diagnostic accuracy. Further analysis revealed that EVs can also be used to distinguish between stage I and stage IIA patients. Although the most profound difference was the increased number of smaller EVs of <100 nm, the authors could define signatures and markers specific for these two stages. The most significantly differentially expressed proteins in stage IIA versus healthy women were P-cadherin and TAZ (transcriptional co-activator with PDZ binding motif), while IGFRβ was the best marker to differentiate between stage I and stage IIA. The levels of TAZ and P-cadherin were reduced in stage IIA compared with healthy controls and also had a negative correlation with numbers of sEVs as expected. The authors concluded that it was unclear why these proteins were found to be decreasing in plasma EVs of patients with more advanced tumours with a higher metastatic potential, postulating that the overall changes in sEV cargo may reflect the “disease state,” which is influenced by tumour cells, in addition to cells of the TME and circulating cells, and does not necessarily recapitulate the expression profile of the tumours or of the EVs that are released from tumour cells. Finally, the authors observed protein-based distinct clusters associated with a high risk of relapse, amongst which the presence of heat shock protein 70 (HSP70) was a predominant feature [176].

A significant study linking elevated levels of circulating EVs with therapy failure and disease progression in BC patients undergoing neoadjuvant chemotherapy demonstrated that (a) the EV concentration was 40-fold higher in breast patients compared to healthy females, (b) the EV concentration increased during therapy, (c) an increased EV concentration before neoadjuvant chemotherapy was associated with therapy failure and (d) an elevated EV concentration after neoadjuvant chemotherapy was associated with a reduced three-year progression-free and overall survival [177].

Another interesting piece of research recently focused on circulating small-sized endothelial microparticles (sEMP) as potential predictive biomarkers in relation to chemotherapy response in patients with BC. Patients with BC treated with neoadjuvant or first-line chemotherapy had baseline and post-treatment circulating sEMP quantified using a flow cytometer approach specifically designed for analysis of small-sized particles (0.1–0.5μm). Median levels of sEMP decreased after chemotherapy (*p* = 0.005). Response to chemotherapy showed a non-significant trend to associate with sEMP response (*p* = 0.056). A sEMP response was observed in 51% of patients and was associated with better overall survival (HR 0.18; 95% CI 0.04–0.87; *p* = 0.02) and progression free survival (HR 0.30; 95% CI 0.09–0.99; *p* = 0.04) in the group of women with metastatic disease [178].

In another study of circulating EV miRNAs from BC patients on neoadjuvant treatment and healthy controls, quantification of 45 EV miRNAs showed that, compared with healthy women, 10 miRNAs in the entire cohort of BC patients, 13 in the subgroup of 211 HER2-positive BC patients and 17 in the subgroup of 224 TNBC patients were significantly deregulated. Plasma levels of 18 EV miRNAs differed between HER2-positive and TNBC subtypes and 9 miRNAs also differed from healthy women. EV miRNAs were significantly associated with clinicopathological and other risk factors, whereas in univariate and multivariate models, miR-155 and miR-301 best predicted pathological complete response (pCR) [179]. Furthermore, in a different study of circulating EV miRNAs derived from patients with TNBC treated with neoadjuvant chemotherapy, non-responsive patients had lower expression levels of miR-185, miR-4283, miR-5008 and miR-3613 and higher expression levels of miR-1302, miR-4715 and miR-3144, whereas the molecular pathways most affected by this combination of miRNAs in non-responders were highly linked to immunosuppressive pathways [180].

In HER2^+^ BC patients, a study of circulating EVs demonstrated an upregulated expression level of serum EV lncRNA SNHG14 (long non-coding-small nucleolar RNA host gene 14) in patients who exhibited resistance to trastuzumab, compared with patients who exhibited a response. In the same study, in vitro knockdown of lncRNA SNHG14 potently promoted trastuzumab-induced cytotoxicity, whereas extracellular lncRNA SNHG14 was able to be incorporated into EVs and transmitted to sensitive cells, thus disseminating trastuzumab resistance. Therefore, lncRNA SNHG14 may be a promising predictive biomarker of response for patients with HER2^+^ BC [181].

Moreover, another study investigating the circulating EV HOTAIR (HOX transcript antisense RNA) expression in BC patients demonstrated that BC patients expressed higher serum EV HOTAIR than healthy individuals, serum EV HOTAIR levels 3 months after surgery were markedly decreased compared with levels before surgery, whereas high expression of EV HOTAIR led to a worse disease-free survival and overall survival. Moreover, in the high-expression neoadjuvant chemotherapy group, six patients achieved a partial response and eight demonstrated stable disease (SD), whereas nine patients achieved a partial response and two SD in the low-expression group. In the low-expression adjuvant tamoxifen group, one patient had a recurrence of BC and another 10 patients exhibited no recurrence, while six showed recurrence and seven had none in the high expression group, thus indicating that serum EV HOTAIR exhibits the potential to be a diagnostic, prognostic and predictive biomarker in BC [182].

In another study, circulating EVs in peripheral blood from BC patients were found to carry the transient receptor potential cation channel subfamily C member 5 (TRPC5), which was previously found to be an essential element for acquired chemoresistance in BC cells. In the present study, the level of circulating EVs carrying TRPC5 (cirExo-TRPC5) was significantly correlated with TRPC5 expression levels in BC tissues and tumour response to chemotherapy. Furthermore, increased cirExo-TRPC5 level after chemotherapy preceded progressive disease (PD) based on imaging examination and strongly predicted acquired chemoresistance [183]. Similarly, ubiquitin carboxyl terminal hydrolase-L1 (UCH-L1), a factor conferring doxorubicin-resistance to BC cells, was found to be highly expressed in circulating EVs from BC patients with poor prognosis due to chemo-resistance [184]. Finally, another study demonstrated that TGFβ1 levels were significantly higher in EVs isolated from the serum of patients with HER2-overexpressing BC who subsequently did not respond to neoadjuvant HER2-targeted drug treatment, compared with those who experienced complete or partial response [185].

Collectively, these early data above support the use of circulating EV proteomic and miRNA analysis for the screening and diagnosis of BC, as well as defining prognosis and response to systemic treatment. One of the main current challenges is the development of accessible, reliable, efficient and low-cost techniques to facilitate detection and quantification of circulating EVs and their BC-specific proteomic and miRNA signatures, in order to allow for their incorporation in daily clinical routines. Furthermore, most data supporting the role of circulating EVs in BC as diagnostic, prognostic and predictive biomarkers, currently originate from small-sized, proof-of-concept studies and ultimately need validation in large scale prospective studies in the neoadjuvant and metastatic setting, as well as epidemiological studies for screening purposes. However, given the potential beneficial implications for patients and clinicians, a ‘liquid biopsy’ strategy on the basis of circulating EVs in BC is certainly worth further exploration and development.

### 6.2. Targeting Extracellular Vesicles to Overcome Therapy Resistance

Several strategies aiming to overcome EV-induced therapeutic resistance in BC are currently under evaluation. These include approaches targeting EV pathways at various biological stages, including EV biogenesis, release and uptake [186] (Figure 4).

Lipid-related pathways such as the conversion of sphingomyelin to ceramide by sphingomyelinases, phosphatidylserine (PS) translocation by protein kinases and cholesterol synthesis are considerably involved in EV production and release [186]. The inhibition of any of these mechanisms may hold promise in preventing EV-mediated therapeutic resistance.

Sphingomyelinase (SMase)-targeting inhibitors, including imipramine and GW4869, are proven to reduce exosome secretion, at varying levels of efficiency in different cancer cells [187]. In particular, GW4869 has been successfully used to block the secretion of exosomes from different BC cell lines, resulting in a reduced proliferation rate and chemosensitisation [188,189]. Recent studies reported that GW4869 treatment restored trastuzumab sensitivity in trastuzumab-resistant BC cells and that incubation with culture medium from trastuzumab-resistant cells treated with GW4869 failed to confer trastuzumab resistance to recipient cells [189,190]. Moreover, Yang et al. showed that inhibition of exosome secretion by GW4869 sensitised breast tumours to immune checkpoint blockade by reducing secreted exosomal PD-L1 [133].

Bisindolylmaleimide-I (BIM1), which targets the ATP-binding site of various protein kinase C isoforms, has been shown to hinder the release of calcium and the externalisation of PS, inhibiting the release of MVs. In addition, it has been reported to potentiate 5-fluorouracil (5-FU)-mediated apoptosis of BC cells [191].

EV release in the microenvironment also involves cytoskeletal reorganisation, modulation of mitochondria-mediated signalling, stimulation of calcium channels and activation of the ESCRT-dependent pathway. In a study by Li et al., it was demonstrated that treatment of BC cells with Y27632, a competitive inhibitor of Rho-associated coiled-coil-containing protein kinase 1/2 (ROCK1/2), previously shown to have a significant impact on BC tumour formation in mice, induced a decrease in secreted MVs [59].

Kosgodage et al. showed that chloramidine (Cl-amidine), a calcium chelator inducing the deimination of cytoskeletal actin involved in MVs biogenesis [59], effectively inhibits EVs release enhancing BC cells response to 5-FU treatment [191].

In 2018, Khan et al. reported that ketotifen, a store-operated calcium channel blocking agent previously demonstrated to have cytotoxic effects on BC and leukaemia cells [192,193,194], inhibited exosome release from BC cells by altering intracellular calcium levels and strengthening the anti-tumour effects of doxorubicin [195].

Sulfisoxazole, an FDA approved antibiotic, reduced the secretion of small EVs in breast adenocarcinoma cell lines by inhibiting components of the ESCRT-dependent pathway, such as ALG-2-interacting protein X (ALIX) and VPS4B, through endothelin receptor type A (a G-protein coupled receptor) targeting and triggering co-localisation of multivesicular endosomes with lysosomes for degradation [196].

Cannabidiol, a phytocannabinoid derived from Cannabis sativa, with proven anti-inflammatory and antioxidant properties, has recently been found to block exosome and microvesicle release, sensitising chemoresistant BC cells to cisplatin [197,198]. Its action is associated with changes in mitochondrial function, including modulation of STAT3 and prohibitin expression, both of which positively regulate cell proliferation [198].

Reduction of EV secretion has also been achieved through inhibition of mortalin, a mitochondrial chaperone protein found in the cytoplasm, endoplasmic reticulum and cytoplasmic vesicles involved in many cellular processes such as mitochondrial biogenesis, intracellular trafficking, cell proliferation, signalling, immortalisation and tumorigenesis [199]. The mortalin inhibitors: MKT-077, dimethyl amiloride (DMA) and omeprazole increased both paclitaxel- and cisplatin-induced apoptosis in BC cell lines. A similar outcome has been observed after exposure of BC cell lines to secretion modification region (SMR)-derived peptides, which fostered reestablishment of complement-mediated cytotoxicity [200,201].

Beyond its activity on mortalin, DMA exerts its inhibitory effect on EV release by interfering with the activity of efflux pumps expressed on acidic vacuoles, such as Na^+^/H^+^ export [202]. In particular, exosome depletion by DMA restored the anti-tumour efficacy of cyclophosphamide, an anticancer drug with cytotoxic and immunological properties [203,204], by inhibiting MDSC functions [204].

The Rab family can also be targeted to inhibit the release of exosomes. shRNA targeting Rab27a reduced local growth of tumour and metastasis of mouse mammary tumour 4T1 cells [205].

Chen et al. reported that D Rhamnose β-hederin (DRβ-H), an active component extracted from the traditional Chinese medicinal plant, Clematis ganpiniana, was effective against BC, owing to its ability to reverse docetaxel resistance through reduction of exosome formation and release from docetaxel-resistant BC cells [206].

A recent study reported that the microenvironmental pH of tumour cells is another key player in the modulation of EV trafficking [207]. At a low pH, the level of EV release and uptake was found to be increased in different types of cancers [208]. Interestingly, data showed that the alkalisation of the TME induced a significant decrease in the number of EVs released by BC cells [208]. Therefore, the inhibition of proton pumps such as V-ATPases and carbonic anhydrase IX (CA IX), might provide a new strategy to block EV release. To date, there is only one CA IX inhibitor (SLC-0111) in Phase Ib/II clinical trials for the treatment of advanced solid tumours [209]. However, several preclinical studies have already reported that CAIX targeting with SLC-0111 has anti-tumour effects in TNBC, increasing response to immune checkpoint blockade and doxorubicin [210,211,212,213,214].

Preventing the uptake of EVs released in the TME could also constitute a good strategy to reduce EV-mediated downstream effects. Therefore, inhibition of clathrin-dependent and clathrin-independent endocytosis pathways, including micropinocytosis, may be a potential therapeutic approach [186].

Methyl-β-cyclodextrin (MβCD), a molecule used to remove cholesterol from membranes, has been reported to diminish EV uptake in BC cells by interfering with lipid raft stability [215]. The same outcome was observed after dynasore treatment, which interfered with EV-mediated invasiveness of cancer cells [216].

The NHE blocker, ethyl isopropyl amiloride (EIPA), currently considered as the most effective and selective inhibitor of macropinocytosis [217], has shown efficacy in BC cells [218].

Rather than targeting EV biogenesis, release or uptake, EVs may also be removed from circulation.

Marleau and colleagues developed a hemofiltration system which decreased systemic HER2^+^ cancer-derived EVs, found to stimulate tumour growth and induce the formation of premetastatic niches [219]. In other studies, antibodies against CD9 and CD63 were used to deplete EVs, decrease metastasis and enhance the therapeutic effect of tamoxifen in mouse models of BC [79,220]. In addition, Gobbo et al. found that an anticancer immune response could be restored by targeting tumour-derived exosomes with a peptide aptamer [221].

All therapeutic approaches listed above present great potential in suppressing the EV-mediated mechanisms of chemoresistance. However, unlike the hemofiltration system, all the other procedures do not specifically target cancer-derived EVs, holding off-target effects which eventually affect physiological communication between cells. Therefore, further work is certainly needed to understand how to specifically target harmful EVs in order to reduce the side effects of these strategies in potential clinical settings.

### 6.3. Extracellular Vesicle-Based Cancer Immunotherapies

Over the past 50 years, BC immunotherapy has attempted to re-tune and modulate the immune system in order to generate novel, targeted treatments for the disease. Immunotherapies hold enormous potential to improve survival in BC, particularly for the subtypes carrying the poorest prognoses, and promising clinical responses have been observed in several ongoing clinical trials [130].

In this regard, tumour-derived EVs carrying tumour-specific and tumour-associated antigens may be reasonably used as sources to stimulate immunity against cancer cells (Figure 5a). In particular, it has been shown that both dendritic- and tumour cell-derived EVs stimulate tumour antigen-specific CD8^+^ cytotoxic T lymphocyte responses, mediating anti-tumour immunity in experimental animal models and human clinical trials in several types of cancer [146,222] (Figure 5a). Interestingly, in other studies, EVs have been genetically engineered for displaying specific monoclonal antibodies on their surface, resulting in novel synthetic multivalent antibodies retargeted exosomes (SMART-Exos) that can simultaneously target immune and cancer cells (Figure 5b). Recently, Shi and colleagues genetically modified HEK293T-derived exosomes, making them express on their surface both anti-human CD3 and anti-human HER2 antibodies, resulting in SMART-Exos dual targeting of T cell-CD3 and BC-associated HER2 receptors. By redirecting and activating cytotoxic T cells toward attacking HER2-expressing BC cells, the designed SMART-Exos exhibited highly potent and specific anti-tumour activity both in vitro and in vivo [223]. A similar approach was used by Cheng et al. who generated SMART-Exos expressing monoclonal antibodies specific for T cells CD3 and cancer cell-associated EGFR, which induced both cross-linking of T cells and EGFR-expressing BC cells, but also elicited potent anti-tumour immunity both in vitro and in vivo [224].

Taghikhani et al. modified breast tumour-derived exosomes (TEX) with miR-155, miR-142, and let-7i, to enhance their immune-stimulating abilities and induce potent dendritic cells, supporting the use of modified TEX as a potential cell-free vaccine for BC treatment [225] (Figure 5a).

In addition, DC-derived exosomes have been proven safe for vaccine delivery in multiple phase I trials in different types of cancers [226,227,228]. Recently, Li et al. developed a novel HER2-specific exosome-T vaccine using polyclonal CD4^+^ T cells armed with exosomes derived from HER2-specific DCs and demonstrated its therapeutic effect in a HER2 antibody therapy-resistant mouse model [229] (Figure 5c). The efficacy of this approach has been further improved by developing a heterologous human/rat HER2-specific exosome-targeted T-cell vaccine using polyclonal CD4^+^ T-cells which take up exosomes released by DC transfected with an adenoviral vector encoding a fusion protein composed of HER2 fragments. This vaccine stimulated enhanced CD4^+^ T-cell responses leading to increased induction of HER2-specific antibodies [230]. Altogether, the approaches described in these preliminary studies may provide a new therapeutic alternative for patients with trastuzumab-resistant HER2^+^ breast tumours.

### 6.4. Extracellular Vesicles as Delivery Vehicles for Therapeutic Agents

A wide range of studies have recently shown the effectiveness of EVs as delivery vectors of different therapeutic agents, including chemotherapeutic drugs, peptides and non-coding RNAs to enhance anti-tumour therapy and reverse drug resistance (Figure 6) [231]. Engineered EVs derived from distinct biosources, including cultured cancer and immune cells, human blood and milk, containing modifications on their membrane making them tissue-specific, have emerged as a potential EV-based cancer therapy [232,233] (Table 3). Compared with some traditional nanomaterials, EVs are biocompatible, biodegradable, have a low toxicity and are non-immunogenic [234]. Moreover, the lipid bilayer membrane of EVs protects the cargo from degradation and the non-targeted cytotoxicity can be reduced due to the presence of transmembrane and membrane anchoring proteins on their surface, facilitating endocytosis and cargo transfer [235].

Below we summarise some of the recent advances in the use of EVs as vehicles for delivering therapeutic agents and improving anti-cancer treatments in BC.

In a study by Liu et al., hyaluronic acid (HA)-functionalised HEK293T-derived extracellular vesicles were used as natural vehicles to efficiently deliver doxorubicin and reverse multi drug resistance (MDR) in BC. In preclinical MDR tumour models, HA-EVs deeply penetrated tumour tissue and effectively transported doxorubicin into tumours, while eliminating doxorubicin’s systemic toxicity [236].

In another study, HEK293T-derived EVs harbouring an EGFR aptamer and loaded with survivin siRNA inhibited BC growth in mice [237].

In a xenograft BC model, Ohno et al. efficiently administered HEK293T-derived EVs modified with the transmembrane domain of PDGFR fused to a GE11 peptide to deliver the cancer suppressor let-7a miRNA to EGFR-expressing BC tissue [238]. Afterwards, Wang et al. successfully enhanced let-7 and VEGF miRNA tumour delivery using DC-derived EVs modified with the AS1411 aptamer, which binds to nucleolin, a protein highly expressed on the membrane of BC cells [239]. In a study by O’Brien et al., MSCs were engineered to secrete EVs enriched with the tumour suppressor miR-379. Systemic administration of cell-free EVs enriched with miR-379 was demonstrated to have a therapeutic effect in BC, mediated, in part, through regulation of cyclooxygenase-2 (COX-2) [240].

The great potential of using MSC-derived EVs as nanovehicles of RNA-based therapeutics has been demonstrated once more by Naseri et al. [241]. Their results indicated that MSC-derived EVs could efficiently deliver LNA-antimiR-142-3p to BC stem-like cells, reducing miR-142-3p and miR-150 expression levels and inhibiting clonogenicity and tumorigenicity [241].

MSC-derived EVs can also be good candidates to carry peptides or recombinant proteins. As reported by Yuan et al. MSC-EV-mediated TRAIL (TNF-related apoptosis inducing ligand) delivery induced pronounced apoptosis and overcame TRAIL resistance in BC cells [242].

The use of EVs as drug carriers in BC has also been widely explored. In a study by Wang et al., EVs derived from proinflammatory macrophages (M1-type RAW264.7) were used as paclitaxel carriers. Beyond the induction of a pro-inflammatory environment, they enhanced the anticancer efficiency of paclitaxel in BC in vitro and in vivo [243]. Similarly, Fan et al. reported an anti-tumour activity of doxorubicin-loaded M1-derived EVs in a BC mouse model [244]. In another study, Li et al. developed a macrophage-derived EV-coated poly (lactic-co-glycolic acid) nanoplatform for targeted chemotherapy of TNBC. To further improve the tumour targetability, the surface of the EVs was modified with a peptide to target the mesenchymal-epithelial transition factor (c-Met), which is overexpressed by TNBC cells. These engineered exosome-coated nanoparticles significantly improved the cellular uptake efficiency and the anti-tumour efficacy of doxorubicin [245].

Tian et al. used EVs isolated from engineered mouse immature DC (imDCs) expressing Lamp2b, a well-characterised exosomal membrane protein, fused to αv integrin-specific iRGD peptide (CRGDKGPDC) to convey doxorubicin to cancer cells. The targeted delivery of doxorubicin to αv integrin-positive BC cells in vitro and to tumour tissues in vivo, led to inhibition of tumour growth without over toxicity [246].

In a similar way, Gong et al. isolated EVs with increased binding to integrin αvβ3 (A15 metalloproteinase-expressing EVs) from monocyte-derived macrophages to deliver doxorubicin and cholesterol-modified miRNA 159 (Cho-miR159) to TNBC cells, both in vitro and in vivo. In vitro, A15-Exo co-loaded with doxorubicin and Cho-miR159 induced synergistic therapeutic effects in a TNBC cell line. In vivo, miR159 and doxorubicin delivery in a vesicular system effectively silenced TCF-7 (transcription factor 7) and exhibited improved anticancer results without adverse effects [247].

In agreement with the recent observation that EVs derived from autologous cancer cells have potential tropism to the TME making them competitive delivery vehicles with enhanced anticancer efficacy [254], Tian et al. developed tumour-cell-derived exosome-camouflaged porous silicon nanoparticles (E-MSNs) as a drug delivery system for indocyanine (ICG) and doxorubicin (ID@E-MSNs). This approach showed synergistic effects of chemotherapy and photothermal therapy against BC [248].

Similarly, Ma et al. used cisplatin-, methotrexate- and doxorubicin-packed tumour-derived EVs to revert BC resistance to chemotherapeutic drugs [250].

Moreover, Kanada et al. demonstrated that tumour-derived microvesicles loaded with a minicircle DNA encoding the prodrug converting enzymes, thymidine kinase (TK)/nitroreductase (NTR), led to the effective killing of BC cells via TK-NTR-mediated conversion of co-delivered prodrugs into active cytotoxic agents [249].

In a study by Zhao et al. autologous BC cell-derived biomimetic nanoparticles (cationic bovine serum albumin (CBSA) conjugated siS100A4 and exosome membrane coated nanoparticles, CBSA/siS100A4@Exosome) were used to deliver siS200A4 to the lung premetastatic niche in a TNBC model. They exhibited outstanding gene-silencing effects that significantly inhibited the growth of malignant BC cells [251].

Another potentially ideal source of EVs could be human red blood cells (RBCs). Usman et al. validated a new strategy to generate large-scale amounts of RBC-derived EVs for the delivery of antisense oligonucleotides (ASOs). In particular, they demonstrated the therapeutic potential of RBC-derived EVs in delivering ASOs that antagonise miR-125b, a well-known oncogenic microRNA, in BC [252].

Finally, biofluid-derived EVs might function as delivery vehicles as shown by Aqil et al. who demonstrated that oral administration of curcumin-loaded milk-derived exosomes enhanced antiproliferative, anti-inflammatory, and anti-tumour activities against multiple cancer cell lines, including BC [253].

## 7. Conclusions

Enormous advances in early detection and targeted treatments have vastly improved BC prognosis, however, therapy failure due to drug resistance remains a challenge. The crosstalk between cancer cells and the TME through EVs is a key mediator of therapy resistance and therefore represents a potential strategy in BC management. Furthermore, the association of EV cargo with drug resistance may provide a useful means of disease monitoring through liquid biopsies in a predictive and prognostic capacity. Additionally, altering EV biogenesis and the manipulation of EV cargo could represent effective novel treatments in BC.

## Figures and Tables

**Figure 1 biomolecules-12-00132-f001:**
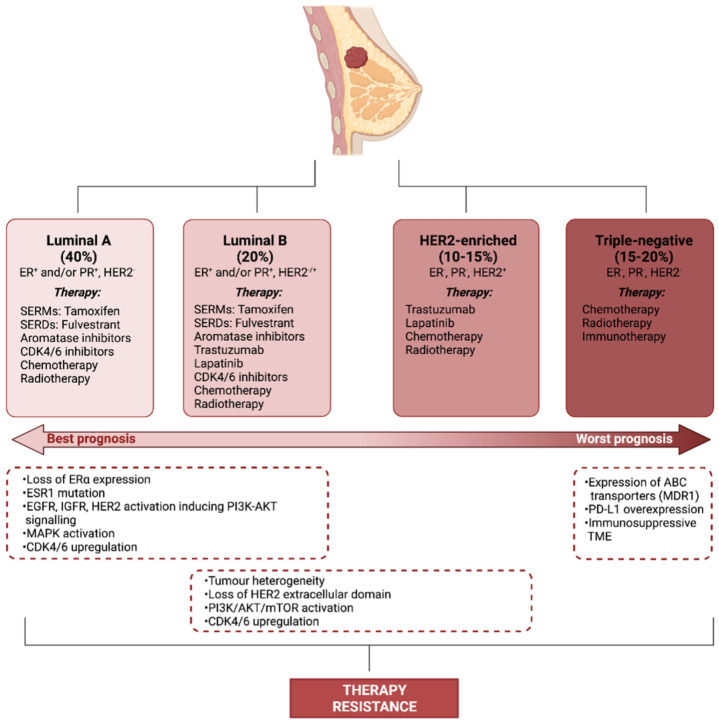
Breast cancer subtypes, standard treatments and mechanisms of therapy resistance. BC can be classified into four main subtypes based on the expression of oestrogen receptor (ER), progesterone receptor (PR) and human epidermal growth factor receptor 2 (HER2). Different treatments are administered based on the molecular markers. Several mechanisms of therapy resistance can arise in the different BC subtypes. (Abbreviations: ABC, ATP-binding cassette; CDK4, cyclin dependent kinase 4; CDK6, cyclin dependent kinase 6; ERα, oestrogen receptor α; EGFR, epidermal growth factor receptor; ESR1, oestrogen receptor 1; IGFR, insulin-like growth factor receptor; MAPK, mitogen-activated protein kinase; mTOR, mammalian target of rapamycin; PD-L1, programmed death ligand 1; PI3K, phosphatidylinositol-3-kinase; SERMS, selective oestrogen receptor modulators; SERDS, selective oestrogen receptor downregulators; TME, tumour microenvironment).

**Figure 2 biomolecules-12-00132-f002:**
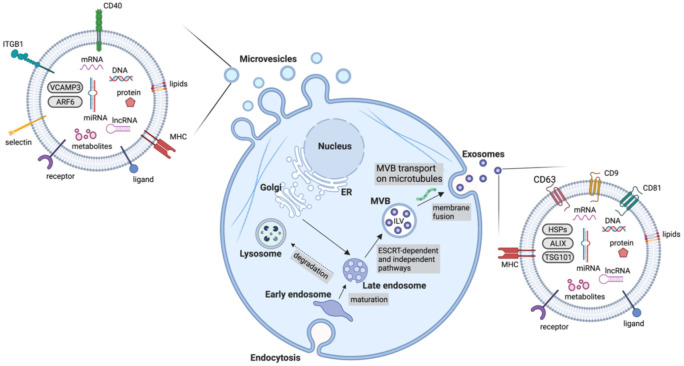
Schematic representation of exosome and microvesicle biogenesis, release and cargo. While microvesicles bud directly from the plasma membrane, exosomes are generated within MVB subpopulations that, upon maturation, fuse with the plasma membrane. Alternative MVB pathways include fusion with lysosomes for degradation. (Abbreviations: ALIX, ALG-2 interacting protein X; ARF6, ADP ribosylation factor 6; ER, endoplasmic reticulum; ESCRT, endosomal sorting complexes required for transport; HSPs, heat shock proteins; ILV, intraluminal vesicles; ITGB1, integrin β1; lncRNA, long non-coding RNA; MHC, major histocompatibility complex; miRNA, microRNA; MVB, multivesicular body; TSG101, tumour susceptibility gene 101; VCAMP3, vesicle-associated membrane protein 3).

**Figure 3 biomolecules-12-00132-f003:**
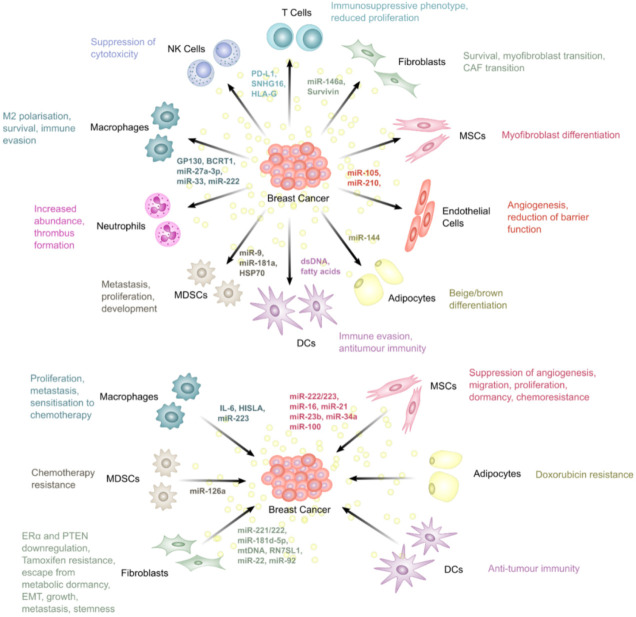
Bidirectional EV communication between BC and the TME. EVs from BC cells contain miRNA, DNA, lipid and protein cargo which is used to alter the behaviour of the TME, promoting therapy resistance. In turn, cells of the TME communicate via EVs which are taken up by BC cells to produce therapy resistance and more aggressive disease. (Abbreviations: BC, breast cancer; BCRT1, breast cancer related transcript 1; CAF, cancer-associated fibroblast; DC, dendritic cell; dsDNA, double-stranded DNA; EV, extracellular vesicle; GP130, glycoprotein 130; HISLA, HIF-1α-stabilising long noncoding RNA; HLA-G, the human leukocyte antigen-G; HSP70, heat shock protein 70; IL-6, interleukin-6; MDSC, myeloid-derived suppressor cell; MSC, mesenchymal stem cell; NK, natural killer; PD-L1, programmed death-ligand 1; SNHG16, small nucleolar RNA host gene 16; TME, tumour microenvironment).

**Figure 4 biomolecules-12-00132-f004:**
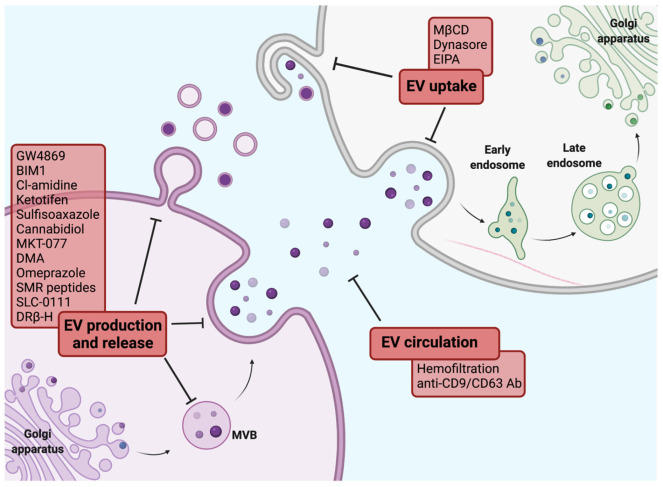
Potential therapeutic interventions targeting EV pathways. The exosome and microvesicle pathways can be targeted at various biological stages including biogenesis, release, circulation and uptake. (Abbreviations: BIM1, Bisindolylmaleimide-I, Cl-amidine, chloramidine; DMA, Dimethyl Amiloride; DRβ-H, D Rhamnose β-hederin; EIPA, ethyl isopropyl amiloride; EV, extracellular vesicles; MβCD, Methyl-β-cyclodextrin; MVB, multivesicular bodies; SMR, secretion modification region).

**Figure 5 biomolecules-12-00132-f005:**
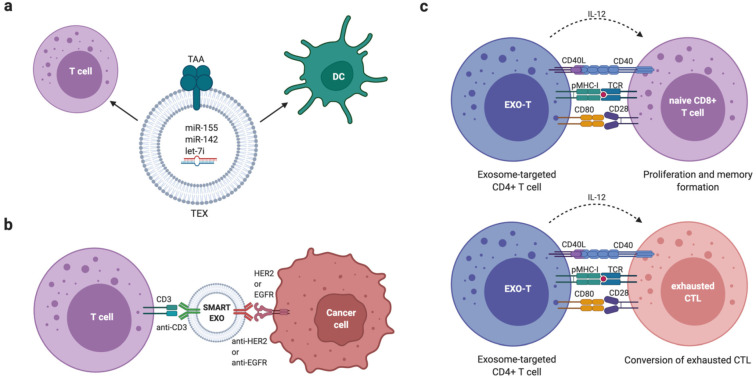
Extracellular Vesicle-Based Cancer Immunotherapies. (**a**) Tumour-derived EVs carrying tumour-specific and tumour-associated antigens (TAAs) and modified breast tumour-derived exosomes (TEX) with miR-155, miR-142, and let-7i, may be reasonably used as sources to stimulate T cells and dendritic cells (DCs) against cancer cells. (**b**) EVs have been genetically engineered for displaying specific monoclonal antibodies on their surface, such as anti-CD3 and anti-HER2 (human epidermal growth factor receptor 2) or anti-EGFR (epidermal growth factor receptor) resulting in novel synthetic multivalent antibodies retargeted exosomes (SMART-EXO) that can simultaneously target immune and cancer cells. (**c**) A novel HER2-specific exosome-T vaccine using polyclonal CD4+ T cells armed with exosomes derived from HER2-specific DCs (EXO-T) has been developed. Distinct three signals derived from novel EXO-T vaccine include (1) exosomal pMHC-I/TCR, (2) exosomal CD80/CD28 and T cell CD40L/CD40 (for T-cell memory formation), and (3) T-cell cytokine IL-2 (for T-cell proliferation). Conversion of exhausted CD8+ CTLs (cytotoxic T lymphocytes) within tumours by EXO-T cells via a T cell CD40L/CD40-activated mTORC1 pathway can also occur.

**Figure 6 biomolecules-12-00132-f006:**
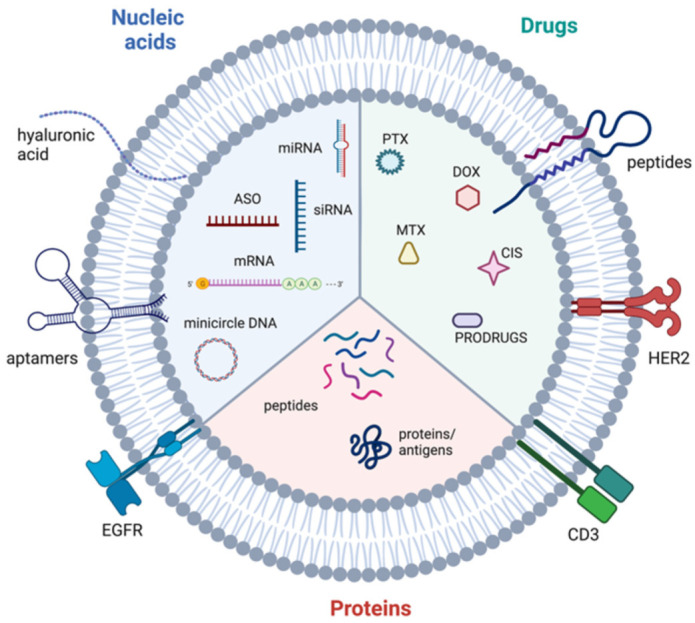
EVs as delivery vehicles. Therapeutic nucleic acids, including DNAs, RNAs and antisense oligonucleotides (ASO), chemotherapeutic drugs, including doxorubicin (DOX), paclitaxel (PTX), cisplatin (CIS) and methotrexate (MTX), along with peptides and proteins can be loaded into EVs. EVs can also be functionalised with proteins presenting targeting abilities and/or anti-tumour effects.

**Table 1 biomolecules-12-00132-t001:** Classification and key features of extracellular vesicles.

Type of EV	Size (nm)	Biogenesis	Main Components
Exosomes	30–150	Early endosomes mature into late endosomes. Through the action of ESCRT machinery, MVBs containing ILVs are formed and fuse with the plasma membrane for release	Tetraspanins (CD9, CD63, CD81), HSPs, MVB biogenesis components (ALIX, TSG101)
Microvesicles	100–1000	Outward budding of the membrane followed by fission through contractile machinery	Cell adhesion molecules (integrins, selectins), Death receptors (CD40), VCAMP3, ARF6
Apoptotic bodies	>1000	Cytoplasmic fragmentation during programmed cell death	Histones, Annexin V, Caspase 3, Phosphatidylserine
Oncosomes	>1000	Cleavage of large cytoplasmic extensions from cell body	Cytoskeleton components (cytokeratin 18), Tetraspanins (CD9, CD81), Cell adhesion molecules (integrins, ICAM, CD44)

Abbreviations: ALIX, ALG-2 interacting protein X; ARF6, ADP ribosylation factor 6; ESCRT, endosomal sorting complexes required for transport; EV, extracellular vesicle; HSPs, heat shock proteins; ILV, intraluminal vesicles; MVB, multivesicular body; TSG101, tumour susceptibility gene; 101; VCAMP3, vesicle-associated membrane protein 3.

**Table 2 biomolecules-12-00132-t002:** Circulating EVs as diagnostic and prognostic biomarkers.

Biological Fluid	Biomarkers	Therapeutic Approaches Involved	Changes Detected	Potential Clinical Application	Refs
Plasma	144 phosphoproteins	n.a.	Increased in BC patients	BC diagnosis	[165]
Plasma	EGFR, EpCAM, HER2	n.a.	Increased in BC patients	BC diagnosis	[166]
Plasma	CD63/EpCAM/mucin 1	n.a.	Increased in BC patients	BC diagnosis	[167]
Plasma	EpCAM	n.a.	Increased in BC patients	BC diagnosis	[168]
Plasma	CD3, CD56, CD2, CD25, CD9, CD44, CD326, CD133/1, CD142, CD45, CD14	Surgery + adjuvant therapy	Increased in BC patients. CD146 and CD45 downregulated 1 month after surgery	BC diagnosis and monitoring after surgical resection and during adjuvant therapy	[169]
Plasma	Del-1	Surgery	Increased in BC patients and returned to almost normal after tumour removal	BC diagnosis and monitoring after surgical resection	[170]
	Claudin 7	n.a.	Increased in BC patients	BC diagnosis	[171]
Urine	miR-21, MMP-1	n.a.	miR-21: decreased in BC patients; MMP1: increased in BC patients	BC diagnosis	[172]
Serum	H19	Surgery	H19 levels increased in BC patients. Median serum EV H19 levels were significantly lower in post-operative than that in the pre-operative setting. EV H19 expression levels were associated with lymph node metastasis, distant metastasis, TNM stages, ER, PR, and HER2 expression	BC diagnosis and monitoring after surgical resection	[173]
Serum	miR-1910-3p	n.a.	Increased in BC patients	BC diagnosis	[174]
Serum	miRNA-21, miRNA-105 miRNA-222	Neoadjuvant therapy	miRNA-21 and 105: increased in metastatic vs non-metastatic patients and healthy controls; miRNA-222 increased in basal-like and in luminal B versus luminal A tumour subtypes. Correlated with clinical and pathological variables such as PR status and Ki67. miRNA-21: during neoadjuvant treatment expression levels directly correlated with tumour size and inversely with Ki67 expression.	BC diagnosis and monitoring disease during neoadjuvant therapy	[175]
Plasma	Ca15-3, CEA, Ca125, HER2, EGFR, PSMA, EpCAM, VEGF	Surgery, chemotherapy	Increased in BC patients. A higher level was significantly associated with PFS	BC diagnosis and monitoring during therapy	[39]
Serum	AnxA2	n.a.	Significantly elevated in TNBC in comparison to ER^+^, HER2^+^. Associated with tumour grade poor overall survival and poor disease-free survival	BC diagnosis	[100]
Plasma	FAK, Fibronectin, P-cadherin, TAZ, IGFRβ, HSP70	Chemotherapy, Radiotherapy, Surgery	FAK, Fibronectin: differentiated BC patients from healthy individuals; P-cadherin, TAZ: reduced in stage IIA; IGFRβ: overexpressed in stage IIA; HSP70: associated with a high risk of relapse	BC diagnosis and monitoring of relapse	[176]
Plasma	EV concentration	Neoadjuvant therapy	Higher in BC patients. EV concentration increased during therapy, an increased EV concentration before neoadjuvant chemotherapy was associated with therapy failure and an elevated EV concentration after neoadjuvant chemotherapy was associated with a reduced three-year progression-free and overall survival	BC diagnosis and monitoring during neoadjuvant therapy	[177]
Plasma	sEMP concentration	Neoadjuvant or first line chemotherapy	Median levels of sEMP decreased after chemotherapy and was associated with better overall survival and progression free survival	BC monitoring during therapy	[178]
Plasma	miR-155, miR-301	Neoadjuvant therapy	predicted pathological complete response	BC monitoring during neoadjuvant therapy	[179]
Blood	miR-185, miR-4283, miR-5008, miR-3613 miR-1302, miR-4715, miR-3144	Neoadjuvant therapy	miR-185, miR-4283, miR-5008, miR-3613: lower expression in non-responsive patients; miR-1302, miR-4715, miR-3144: higher expression in non-responsive patients	TNBC monitoring during neoadjuvant therapy	[180]
Serum	SNHG14	Trastuzumab	upregulated in HER2^+^ patients who exhibited resistance to Trastuzumab	HER2^+^ BC monitoring during Trastuzumab treatment	[181]
Serum	HOTAIR	Surgery, neoadjuvant therapy	Decreased after surgery	BC monitoring during therapy	[182]
Plasma	TRPC	Chemotherapy	Increased cirExo-TRPC5 level after chemotherapy preceded PD based on imaging examination and strongly predicted acquired chemoresistance	BC monitoring during therapy	[183]
Serum	UCH-L	Chemotherapy	Highly expressed in BC patients with poor prognosis due to chemo-resistance	BC monitoring during therapy	[184]
Serum	TGFβ1	Neoadjuvant therapy	Increased in patients with HER2-overexpressing BC who did not respond to neoadjuvant HER2-targeted drug treatment	BC monitoring during neoadjuvant therapy	[185]

Abbreviations: AnxA2, annexin A2; BC, breast cancer; Ca15-3, carcinoma antigen 15-3, CEA, carcinoembryonic antigen, Ca125, cancer antigen 125; Del-1, developmental endothelial locus-1 protein; EGFR, epidermal growth factor receptor; EpCAM, epithelial cell adhesion molecule; ER, oestrogen receptor; EV, extracellular vesicle; FAK, focal adhesion kinase; HER2, human epidermal growth factor receptor 2; HOTAIR, HOX transcript antisense RNA; HSP70, heat shock protein 70; MMP-1, matrix metalloproteinase-1; n.a., not applicable; PD, progressive disease; PFS, progression-free survival; PR, progesterone receptor; PSMA, prostate-specific membrane antigen; sEMP, small-sized endothelial microparticles; SNHG14, long non-coding-small nucleolar RNA host gene 14; TAZ, transcriptional co-activator with PDZ binding motif; TGFβ1, transforming growth factor β1; TNBC, triple-negative breast cancer; TRPC, transient receptor potential cation channel subfamily C member 5; UCH-L, ubiquitin carboxyl terminal hydrolase-L1; VEGF, vascular endothelial growth factor.

**Table 3 biomolecules-12-00132-t003:** List of main extracellular vesicles used as delivery vehicles for therapeutic agents.

Source of EVs	Cargo	Outcomes	Refs
HEK293T	Doxorubicin	Inhibited MDR tumour growth and extended animal survival times	[236]
	EGFR aptamer + Survivin siRNA	Inhibited BC growth in mice	[237]
	Transmembrane domain of PDGFR fused to the GE11 peptide + let-7a miRNA	Inhibited tumour development in vitro and in vivo	[238]
DC	AS1411 aptamer + let-7 and VEGF miRNA	High anti-tumour activity in vitro and in vivo	[239]
MSC	miR-379	Therapeutic effect in vitro and in vivo, mediated, in part, through regulation of COX-2	[240]
	LNA-antimiR-142-3p	Reduced miR-142-3p and miR-150 expression levels and inhibited clonogenicity and tumorigenicity of BC stem-like cells	[241]
	TRAIL	Induced pronounced apoptosis and overcame TRAIL resistance in BC cells	[242]
M1-type RAW264.7	Paclitaxel	Enhanced anticancer efficiency of paclitaxel in BC in vitro and in vivo	[243]
	Doxorubicin	Enhanced anti-tumour activity in a BC mouse model	[244]
	poly(lactic-co-glycolic acid) + anti-c-Met peptide	Improved the cellular uptake efficiency and the anti-tumour efficacy of doxorubicin in TNBC	[245]
imDCs	Lamp2b fused to CRGDKGPDC peptide	Inhibited tumour growth without over toxicity	[246]
THP1	Doxorubicin + Cho-miR159	Improved anticancer results in vivo and in vitro in TNBC	[247]
4T1	Indocyanine + Doxorubicin	Showed synergistic effects of chemotherapy and photothermal therapy against BC	[248]
	TK/NTR minicircle DNA	Effective killing of BC cells	[249]
MCF7	Cisplatin + Methotrexate + Doxorubicin	Reverted BC resistance to chemotherapeutic drugs	[250]
Autologous BC cells	siS200A4	Targeted lung premetastatic niche in a TNBC model	[251]
RBCs	ASOs against miR-125b	Efficient genome editing in BC cells in vitro and in vivo, without any observable cytotoxicity	[252]
Milk	Curcumin	Enhanced antiproliferative, anti-inflammatory, and anti-tumour activities	[253]

Abbreviations: ASO, antisense oligonucleotides; BC, breast cancer; COX-2, cyclooxygenase-2; DC, dendritic cells; EGFR, epidermal growth factor receptor; MDR, multidrug resistant; NTR, nitroreductase; PDGFR, platelet-derived growth factor receptor; RBC, red blood cells; TK, thymidine kinase; TNBC, triple-negative breast cancer; TRAIL, TNF-related apoptosis inducing ligand; VEGF, vascular endothelial growth factor.

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
