# Peer review of "Extracellular Vesicles as Mediators of Therapy Resistance in the Breast Cancer Microenvironment"

_biomolecules, 2022, doi:10.3390/biom12010132_

Round 1

Reviewer 1 Report

Detailed and comprehensive, this review introduces the reader to the various types of cancers under the breast cancer umbrella. Understanding how different breast cancer cell signatures contribute to unique EV signatures is important in using EVs as both diagnostic/prognostic liquid biopsies, as targets for personalized cancer therapy, and potential mediators of disease intervention.

Author Response

Detailed and comprehensive, this review introduces the reader to the various types of cancers under the breast cancer umbrella. Understanding how different breast cancer cell signatures contribute to unique EV signatures is important in using EVs as both diagnostic/prognostic liquid biopsies, as targets for personalized cancer therapy, and potential mediators of disease intervention.

We would like to thank this Reviewer for their interest in our study and for acknowledging its quality and potential impact of our review article.

Reviewer 2 Report

The introduction provides a detailed overview of therapy resistance in breast cancers (BC) and stresses the relevance of tumor subtypes in BC. None of the currently most important forms of therapy are left out. This has textbook quality and provides a concise background for future molecular biologists. It still feels a long way to the actual topic of the review article (extracellular vesicles). This is followed by a much less detailed chapter on the role of the tumor microenvironment - one may wonder why this is described in so much less detail, compared to the introduction of therapies (and resistance) as the TME is at least as relevant for the main topic (vesicles) as the resistance mechanisms. In short, while the intro on therapy failures feels almost too detailed, there is no comparable intro concerning the TME. And the last bits of this mini-"chapter" already lead over to the vesicles. 

At the beginning of the introduction of extracellular vesicles or exosomes, the style of this review article radically changes. It immediately plunges into many molecular details such as the transport or escort mechanisms, with much less of an introductory character (and is also a bit more difficult to read). At this point, the text becomes significantly more difficult to read (and is less "entertaining") than anything before, especially since there is suddenly a flood of abbreviations that challenge the reader's mind (and memory). I don't know how this could be softened, though, as the information content is very dense. It is good and necessary that this is partly illustrated by figures, which are also of good quality. There could be even more figures, in my opinion, to illustrate the dense amount of molecular processes. What could help, for example, are figures illustrating the different types of EVs side by side, for better comparison and boosting the reader's comprehension. Also, size comparisons - mentioned several times throughout the text - could be illustrated in this way. 

One particular bit of information, however, is missing, especially after the different types of EVs have been mentioned: How are these different EVs detected in practice? What are the most commonly used methods to enrich and isolate, which methods are used to identify different types of EVs? This would also be particularly relevant and important as an introduction and necessary background to chapter 6- the clinical applications of EV research. Which methods are used in liquid biopsy testing, specifically targeting Vs, and their content - over just measuring free circulating DNA, for example.  Methods are occasionally mentioned, in passing, throughout chapter 6, but there is no overview chapter or subchapter, that may help the reader to get an overview of the most relevant approaches. Related to this, it may also be interesting to highlight from which different body liquids (like the serum,  CSF, urine, etc) EVs can be successfully extracted and analyzed (also, how this happens). This would be part of providing a comprehensive overview of the entire field, and again highly important for making more sense of chapter 6, in particular.  Currently, in chapter 6, there is a long list of studies focusing on various miRNAs, transferred via EVs, that may be relevant as predictive and diagnostic biomarkers in BC, which is also a bit more difficult to read and understand if the background is not introduced - at least briefly.  Almost every other conceivable aspect of the EV field is mentioned or outlined in much detail in this comprehensive review article - except the diagnostic technologies that are emerging. I think this at least as relevant and interesting as the spectrum of human cell types that are targeted and affected by EVs  - and should at least be mentioned. This may add a page or two to the review; space that could be saved elsewhere, if necessary. 

Another overview-like (sub)chapter that may help could relate to the major classes of molecules contained in the EVs, along with a brief overview of their main molecular functions (or concepts) - and their cellular impact. This does not need to be lengthy, as examples are provided throughout the rest of the text. This is partly covered by Fig. 3, but has no clear equivalent in the text. This could also bridge the gap between introducing the different types of EVs, to the lengthy chapter describing which cell types within the TME may be recipients of EV-mediated messages (starting from line 255). 

The focus and main interest of the authors become rather clear, considering the space given to the different cell types found in the TME, which may be affected by EVs (or their source) and their potential messages. There is nothing wrong with that (everyone has a main area of expertise), but adding some information on the topics mentioned above would still expand the scope and usefulness of the entire review (in my humble opinion). Naturally, the function of miRNAs as the main content of many EVs should be explained in brief (for those not familiar with this). 

Currently, the main portion of the review (chapter 5) also serves as a very detailed introduction to the nature and composition (and the functions) of the TME, with EVs as a central line of communication. That's fine, and appreciated, but it could also be highlighted in the title, to indicate the main tendencies of what is discussed here. 

The discussion on lymphocytes (and immune checkpoint blockade) in connection with EVs is particularly relevant, considering the current hype around immunotherapies. Stressing this content, and highlighting this in the abstract more prominently, could be considered a strategy to boost the impact of this review article in the field. This is not just a "journalistic" trick but would be a necessary focus on the connection between immune checkpoint blockade and EV-mediated mechanisms, also to the emergence of an immunosuppressive TME as a whole which severely affects the impact of immune checkpoint inhibitors on the majority of patients. Given these clinical numbers, an even highlighted focus on EVs in immune evasion appears justified to me. One question would also be interesting: how are immune cells affected b the activities of stromal cells within the TME, as a major mechanism generating an immunosuppressive environment? And would EVs have any role in it? (Very likely).  This is a rapidly emerging field that is currently attracting a lot of attention. I am not an expert in EVs, but maybe there is already published work published out there, that could be included?  

Chapter 6 is, in my opinion, a bit difficult to read, as it provides many many different studies (always introduced as "a recent study..." or "in another study"..) that somehow are difficult to put together into a bigger picture. The large number of "recent studies" also make the reading of this chapter a bit difficult and tedious. This is a pity, considering the high quality of literature research displayed here. It may be helpful to summarize some of these applications in a table? While similar interconnections between findings are nicely highlighted in Fig 3 (highlighting the cellular connections), no such overview is provided for chapter 6. Figure 4 comes close but is focused on the potential interventions that could be utilized in clinical practice. 

Similar issues apply for sub-chapter 6.2, summarizing a large number of possible interventions - although here, the existence of Fig. 4 is really helpful to provide an overview to the reader. But apart from Fig. 4, it is still a bit tedious to read through long lists of compounds, and their effects. Maybe also here, a table that really summarizes the main effects observed, could be really helpful. Also for sub-chapters 6.3 and 6.4, although much shorter, tables may be useful to summarize the main messages (and especially helpful for lazy readers that mainly look for this kind of quick overview). 

small things

line 358: explain what are MRC5 cells

Author Response

We would like to thank this Reviewer for their interest in our report. We believe their comments have helped us improve the present manuscript substantially. We believe we have now addressed this reviewer’s concerns. Please see below.

  1. The introduction provides a detailed overview of therapy resistance in breast cancers (BC) and stresses the relevance of tumor subtypes in BC. None of the currently most important forms of therapy are left out. This has textbook quality and provides a concise background for future molecular biologists. It still feels a long way to the actual topic of the review article (extracellular vesicles). This is followed by a much less detailed chapter on the role of the tumor microenvironment - one may wonder why this is described in so much less detail, compared to the introduction of therapies (and resistance) as the TME is at least as relevant for the main topic (vesicles) as the resistance mechanisms. In short, while the intro on therapy failures feels almost too detailed, there is no comparable intro concerning the TME. And the last bits of this mini-"chapter" already lead over to the vesicles.

We would like to thank the Reviewer for this observation. The main focus of the Review is to describe the implication of EV-specific inter-cellular crosstalk in breast cancer therapeutic resistance. As all the different cell types located in the TME have been extensively described in chapter 5 of the manuscript and the role of the TME in breast cancer progression and therapeutic failure has been deeply described in several previously published papers (which we have now referenced in our article), we believe that such information may not be relevant for the main purpose of this Review. However, following your comment, we have now added some more essential information about the role of TME in breast cancer progression and therapeutic resistance (see lines 148-163.)

  1. At the beginning of the introduction of extracellular vesicles or exosomes, the style of this review article radically changes. It immediately plunges into many molecular details such as the transport or escort mechanisms, with much less of an introductory character (and is also a bit more difficult to read). At this point, the text becomes significantly more difficult to read (and is less "entertaining") than anything before, especially since there is suddenly a flood of abbreviations that challenge the reader's mind (and memory). I don't know how this could be softened, though, as the information content is very dense. It is good and necessary that this is partly illustrated by figures, which are also of good quality. There could be even more figures, in my opinion, to illustrate the dense amount of molecular processes. What could help, for example, are figures illustrating the different types of EVs side by side, for better comparison and boosting the reader's comprehension. Also, size comparisons - mentioned several times throughout the text - could be illustrated in this way.

Following your suggestion, we have now included Table 1 summarizing the different types of EVs side by side. We believe that this is definitely boosting the reader’s comprehension.

  1. One particular bit of information, however, is missing, especially after the different types of EVs have been mentioned: How are these different EVs detected in practice? What are the most commonly used methods to enrich and isolate, which methods are used to identify different types of EVs? This would also be particularly relevant and important as an introduction and necessary background to chapter 6- the clinical applications of EV research. Which methods are used in liquid biopsy testing, specifically targeting Vs, and their content - over just measuring free circulating DNA, for example. Methods are occasionally mentioned, in passing, throughout chapter 6, but there is no overview chapter or subchapter, that may help the reader to get an overview of the most relevant approaches. Related to this, it may also be interesting to highlight from which different body liquids (like the serum,  CSF, urine, etc) EVs can be successfully extracted and analyzed (also, how this happens). This would be part of providing a comprehensive overview of the entire field, and again highly important for making more sense of chapter 6, in particular.  Currently, in chapter 6, there is a long list of studies focusing on various miRNAs, transferred via EVs, that may be relevant as predictive and diagnostic biomarkers in BC, which is also a bit more difficult to read and understand if the background is not introduced - at least briefly.  Almost every other conceivable aspect of the EV field is mentioned or outlined in much detail in this comprehensive review article - except the diagnostic technologies that are emerging. I think this at least as relevant and interesting as the spectrum of human cell types that are targeted and affected by EVs  - and should at least be mentioned. This may add a page or two to the review; space that could be saved elsewhere, if necessary.

Following your observation, we have included section 4.3, discussing how EVs are enriched and analysed to give the necessary background information to chapters 5 and 6. We also included information about the relevance of EVs in liquid biopsies in section 4.4 (see lines 303-315).

  1. Another overview-like (sub)chapter that may help could relate to the major classes of molecules contained in the EVs, along with a brief overview of their main molecular functions (or concepts) - and their cellular impact. This does not need to be lengthy, as examples are provided throughout the rest of the text. This is partly covered by Fig. 3, but has no clear equivalent in the text. This could also bridge the gap between introducing the different types of EVs, to the lengthy chapter describing which cell types within the TME may be recipients of EV-mediated messages (starting from line 255). The focus and main interest of the authors become rather clear, considering the space given to the different cell types found in the TME, which may be affected by EVs (or their source) and their potential messages. There is nothing wrong with that (everyone has a main area of expertise), but adding some information on the topics mentioned above would still expand the scope and usefulness of the entire review (in my humble opinion). Naturally, the function of miRNAs as the main content of many EVs should be explained in brief (for those not familiar with this).

Following your observation, we have now included section 4.4 in order to give information on the kinds of molecules contained within EVs and a brief description of their impact on recipient cells. Additionally, we have briefly discussed the function of miRNAs as background for those unfamiliar with miRNAs.

  1. Currently, the main portion of the review (chapter 5) also serves as a very detailed introduction to the nature and composition (and the functions) of the TME, with EVs as a central line of communication. That's fine, and appreciated, but it could also be highlighted in the title, to indicate the main tendencies of what is discussed here.

Thank you for pointing this out. We agree with this Reviewer on the title and have now modified it.

  1. The discussion on lymphocytes (and immune checkpoint blockade) in connection with EVs is particularly relevant, considering the current hype around immunotherapies. Stressing this content, and highlighting this in the abstract more prominently, could be considered a strategy to boost the impact of this review article in the field. This is not just a "journalistic" trick but would be a necessary focus on the connection between immune checkpoint blockade and EV-mediated mechanisms, also to the emergence of an immunosuppressive TME as a whole which severely affects the impact of immune checkpoint inhibitors on the majority of patients. Given these clinical numbers, an even highlighted focus on EVs in immune evasion appears justified to me. One question would also be interesting: how are immune cells affected b the activities of stromal cells within the TME, as a major mechanism generating an immunosuppressive environment? And would EVs have any role in it? (Very likely). This is a rapidly emerging field that is currently attracting a lot of attention. I am not an expert in EVs, but maybe there is already published work published out there, that could be included? 

We would like to thank you for this observation and suggestion. We have now addressed this issue, highlighting in the abstract the implication of EVs in the failure of novel immunotherapeutic approaches (see lines 11-12).

As rightly mentioned by the Reviewer, the impact of the TME in the regulation of the immunosuppressive phenotype is currently at the forefront of cancer research, considering also the great implication this can have on the development of effective new immunotherapy approaches. Indeed, many papers and reviews have been published in the past few years highlighting the central role played by stromal cells in the creation of an immune-tolerant TME.  The role of EVs in the immunosuppressive TME is a novel and rapidly growing field. Although most of the research is now focused on the impact of BC-derived EVs on immune cells, few papers have been published exploring the effect of stromal-derived EVs on the immune microenvironment and immunotherapeutic resistance. As we reported in our manuscript, MDSC-derived EVs have been described as key mediators in the recruitment of immunosuppressive cells (see lines 668-670). However, other stromal cell types can take part in this process. CAF-derived EVs can regulate PD-L1 expression and mediate therapeutic resistance (see lines 613-614) and it has been shown that after treatment with CAF-derived EVs, BC cells express higher PD-L1, which significantly impairs T cells proliferation and anti-tumour activity (see lines 614-616). Also, MSC-derived EVs can cause differentiation of monocytic myeloid-derived suppressor cells into highly immunosuppressive type 2 -polarised macrophages at tumour site (see lines 544-546).

  1. Chapter 6 is, in my opinion, a bit difficult to read, as it provides many many different studies (always introduced as "a recent study..." or "in another study"..) that somehow are difficult to put together into a bigger picture. The large number of "recent studies" also make the reading of this chapter a bit difficult and tedious. This is a pity, considering the high quality of literature research displayed here. It may be helpful to summarize some of these applications in a table? While similar interconnections between findings are nicely highlighted in Fig 3 (highlighting the cellular connections), no such overview is provided for chapter 6. Figure 4 comes close but is focused on the potential interventions that could be utilized in clinical practice.

Following your suggestion, we have now included Table 2 in the manuscript summarizing all the studies reporting the use of circulating EVs as clinical biomarkers.

  1. Similar issues apply for sub-chapter 6.2, summarizing a large number of possible interventions - although here, the existence of Fig. 4 is really helpful to provide an overview to the reader. But apart from Fig. 4, it is still a bit tedious to read through long lists of compounds, and their effects. Maybe also here, a table that really summarizes the main effects observed, could be really helpful. Also for sub-chapters 6.3 and 6.4, although much shorter, tables may be useful to summarize the main messages (and especially helpful for lazy readers that mainly look for this kind of quick overview).

We thank the Reviewer for pointing this out. Although we believe that Figure 4 summarizes the main message deriving from sub-chapter 6.2, we agree with that sub-chapters 6.3 and 6.4 may be a bit tedious and confusing. Therefore, we have now included Figure 5 and Table 3 in the Manuscript, which summarize, respectively, the role of EVs in cancer immunotherapies and in the delivery of therapeutic agents.

small things

line 358: explain what are MRC5 cells

We would like to thank the reviewer for noticing this. Following your suggestion, we have now included this information in our manuscript.